

# Biomass burning events measured by lidars in EARLINET - Part 2: Optical properties investigation.

Mariana Adam[1], Iwona S. Stachlewska[2], Lucia Mona[3], Nikolaos Papagiannopoulos[3], Juan Antonio Bravo-Aranda[4], Michaël Sicard[5,6], Doina Nicolae[1], Livio Belegante[1], Lucja Janicka[2], Dominika Szczepanik[2], Maria Mylonaki[7], Christina-Anna Papanikolaou[7], Nikolaos Siomos[8,16], Kalliopi Artemis Voudouri[8], Luca Alados-Arboledas[4], Arnoud Apituley[9], Ina Mattis[10], Anatoli Chaikovsky[11], Constantino Muñoz-Porcar[5], Aleksander Pietruczuk[12], Daniele Bortoli[13], Holger Baars[14], Ivan Grigorov[15], Zahary Peshev[15]

[1]National Institute for R&D in Optoelectronics, Magurele, 077225, Romania
[2]Faculty of Physics, University of Warsaw, 02-093, Warsaw, Poland
[3]Consiglio Nazionale delle Ricerche - Istituto di Metodologie per l'Analisi Ambientale (CNR-IMAA), C.da S.Loja. Tito Scalo (PZ), Italy
[4]Andalusian Institute for Earth System Research, Department of Applied Physics, University of Granada, Granada, 18071, Spain
[5]Remote Sensing Laboratory/CommSensLab, Universitat Politecnica de Catalunya, Barcelona, 08034, Spain
[6]Ciencies i Tecnologies de l'Espai - Centre de Recerca de l'Aeronautica i de l'Espai/Institut d'Estudis Espacials de Catalunya (CTE-CRAE/IEEC), Universitat Politecnica de Catalunya, Barcelona, 08034, Spain
[7]National Technical University of Athens, Department of Physics, Athens, 15780, Greece
[8]Laboratory of Atmospheric Physics, Aristotle University of Thessaloniki, Thessaloniki, 54124, Greece
[9]KNMI – Royal Netherlands Meteorological Institute, De Bilt, 3731, the Netherlands
[10]Deutscher Wetterdienst, Meteorologisches Observatorium Hohenpeißenberg, Hohenpeißenberg, 82383 Germany
[11]Institute of Physics, NAS of Belarus, Minsk, 220072, Belarus
[12]Institute of Geophysics, Polish Academy of Sciences, Warsaw, 01-452, Poland
[13]Earth Sciences Institute, Physics Department, University of Évora, Évora, 7000, Portugal
[14]Leibniz Institute for Tropospheric Research, Leipzig, 04318, Germany
[15]Institute of Electronics, Bulgarian Academy of Sciences, 1784, Sofia, Bulgaria
[16]Institute for Astronomy, Astrophysics, Space Applications and Remote Sensing, National Observatory of Athens, Athens, 15236, Greece

*Correspondence to*: Mariana Adam (mariana.adam@inoe.ro)

**Abstract.**

Biomass burning episodes measured at 14 stations of the European Aerosol Research Lidar Network (EARLINET) over 2008-2017 were analysed using the methodology described in "Biomass burning events measured by lidars in EARLINET - Part 1: Data analysis methodology" (Adam et al., 2020, this issue). The smoke layers were identified in lidar optical properties profiles. A number of 795 layers for which we measured at least one intensive parameter was analysed. These layers were geographically distributed as follows: 399 layers observed in South-East Europe, 119 layers observed in South-West Europe, 243 layers observed in North-East Europe, and 34 layers observed in Central Europe. The mean layer intensive parameters are discussed following two research directions: (I) the long-range transport of smoke particles from North America, and (II) the



smoke properties (fresh versus aged), separating the smoke events into four continental source regions (European, North American, African, Asian or a mixture of two), based on back trajectory analysis. The smoke detected in Central Europe (Cabauw, Leipzig, and Hohenpeißenberg) was mostly transported from North America (87% of fires). In North-East Europe (Belsk, Minsk, Warsaw) smoke advected mostly from Eastern Europe (Ukraine and Russia), but there was a significant contribution (31%) from North America. In South-West Europe (Barcelona, Evora, Granada) smoke originated mainly from the Iberian Peninsula and North Africa (while 9% were originating in North America). In the South-East Europe (Athens, Bucharest, Potenza, Sofia, Thessaloniki) the origin of the smoke was mostly local (only 3% represented North America smoke). The following features, correlated with the increased smoke travel time (corresponding to aging) were found: the colour ratio of the lidar ratio (i.e., the ratio of the lidar ratio at 532 nm to the lidar ratio at 355 nm) and the colour ratio of the backscatter Ångström exponent (i.e., the ratio of the backscatter-related Angstrom exponent for the pair 532 nm – 1064 nm to the one for the pair 355 nm – 532 nm) increase, while the extinction Ångström exponent and the colour ratio of the particle depolarization ratio (i.e., the ratio of the particle linear depolarization ratio at 532 nm to the particle depolarization ratio at 355 nm) decrease. The smoke originating from all continental regions can be characterized on average as aged smoke, with a very few exceptions. In general, the long range transported smoke shows higher lidar ratio and lower depolarization ratio compared to the local smoke.

## 1 Introduction

The biomass burning (BB) context was given in Adam et al., 2020 (Part 1, this ACP issue). Therein, the information on BB was reviewed, its importance and role on radiative transfer, air quality and human health, were highlighted, and an overview of the fire monitoring perspective was discussed.

There is a direct link between climate change and forest wildfires. The European Union reports of fires occurrence over Europe (http://effis.jrc.ec.europa.eu/reports-and-publications/annual-fire-reports, last access 13 July 2021) indicate that the climate change induces an increase in the number of fires. Flannigan et al. (2000) modelled the climate change impact, demonstrating an increase of forest wildfire activity. Carvalho et al. (2011) modelled the impact of forest fires in a changing climate on air quality (a case study on Portugal) showing a strong impact on ozone and PM10 (particulate matter with size diameter below 10 μm). One of the current challenges is in evaluating accurately the role of BB in climate change. Besides the BB impacts on climate change, Keywood et al. (2013) describe the impacts of climate change on BB (e.g., fire severity, increase of fuel consumption). The authors state that, based on the BB impact on air pollution, climate, poverty, security, food supply and biodiversity, a more effective control of the fires is needed, along with continuous and improved monitoring.

EARLINET (European Aerosol Research Lidar Network; https://www.earlinet.org/ last access: 10 July 2021; e.g., Pappalardo et al., 2014) provides high temporal and spatial resolution ground-based aerosol measurements, and represents a valuable tool for smoke monitoring. EARLINET is part of the Aerosol Cloud and Trace Gases Research Infrastructure (ACTRIS) (https://actris.eu, last access: 13 July 2021). There are numerous studies describing various BB events over Europe, most of





them focusing on the optical properties of either fresh/local aerosol (e.g. Balis et al., 2003; Alados-Arboledas et al., 2011; Sicard et al., 2012; Nicolae et al., 2013; Stachlewska et al. 2017a,b; Osborne et al., 2019) or aged/long range transported aerosol (Wandinger et al., 2002; Mattis et al., 2003; Müller et al., 2005; Ancellet et al., 2016; Ortiz-Amezcua et al., 2017; Haarig et a., 2018; Stachlewska et al., 2018; Vaughan et al., 2018; Hu et al., 2019; Sicard et al., 2019, Baars et al., 2019).

The aim of this study is to find specific features of the smoke originating from North America and investigate different continental origin of the smoke for each of the four considered geographical regions. The smoke origin is assessed by backtrajectory analyses and the FIRMS product. The analysis is made using intensive parameters (referred to as IPs), which are independent of the aerosol load and are solely aerosol type dependent. This paper presents the Part 2 of investigation of biomass burning episodes as measured by EARLINET, and it focuses on results interpretation. Part 1 (Adam et al, 2020)

described in detail the methodology used to analyse lidar data. Nonetheless, a short overview of the methodology is given in Section 2. In Section 3, we analyse the results for the smoke originating in North America. In Section 4, we focus on results from four European geographical regions, with different continental smoke origin. In Section 5, we provide the summary and conclusions. A list of acronyms used in the current work is given in Appendix A. The location of the EARLINET stations along with the chosen geographical regions are given in Appendix B.

## 2 Methodology

The methodology steps are shown in Fig. S1 (Fig. 2 in Adam et al., 2020). The input for the analysis is the EARLINET/ACTRIS so-called backscatter (b) and extinction (e) files providing the vertical profiles of particle backscatter coefficient, particle extinction coefficient, and particle linear depolarization ratio (when available). In general, for most of the stations the range resolution of profiles is 3.75 m for backscatter coefficients and 60 m for extinction coefficients and the

profiles are averages of 1 h (i.e., various resolutions were used by the stations; Adam et al., 2020). The files are allocated by the stations to the Forest Fire category in the EARLINET/ACTRIS database when an investigation at the station level highlighted the potential presence of smoke layer. The aerosol layer assignment is made manually by the EARLINET stations and it is typically made by means of investigation of intensive parameters (Ångström exponent, lidar ratios, linear particle depolarization ratio, etc), model outputs, backward trajectory analyses, and ancillary instruments data if available. Data are

quality assured following the EARLINET Quality Check (QC) procedures. Most of the data used for this paper are the EARLINET data reported in Forest Fire category labelled as Level 2 data, where 2341 files out of 3589 files (input data) were compliant with all the QC v2.0, at the date of 23 April 2019 (Adam et al., 2020).

Additional data check procedures were applied for the specific purposes and analysis, as described in detail Part 1, here recalled in short. i) For the analysis, a distinct peak in signal amplitude well above the SNR was considered as essential for a layer

identification. ii) For identifying the layer(s) affected by smoke a ten days backtrajectory was computed per each layer using the Hybrid Single-Particle Lagrangian Integrated Trajectory model (HYSPLIT; Stein et al., 2015; Rolph et al., 2017). The meteorological model applied was the Global Data Assimilation System (GDAS), with 0.5° resolution. The identification of



the smoke layers was assessed based on the hypothesis of an existing fire within 100 km and ± 1 h from the location and time of the air mass, respectively. The location of the fires was provided by the Fire Information for Resource Management System (FIRMS) (https://firms.modaps.eosdis.nasa.gov/, last access: 13 July 2021) that uses satellite observations of the Moderate Resolution Imaging Spectroradiometer (MODIS) onboard the Aqua and Terra satellites (Davies et al., 2009). In the current

study, a fire was defined by a specific location (given by latitude and longitude) and a specific time. According to MODIS, latitude and longitude are the middle points of 1 km grid (centre of 1 km fire pixel) but not necessarily the actual location of the fire as one or more fires can be detected within the 1 km pixel. The uncertainties related with the Hysplit backtrajectories or the FIRMS database were not considered. iii) The layer's mean optical properties were calculated only for sufficient signal to noise ratio (SNR ≥ 2) and the number of available data in the layer being ≥ 90% (Adam et al., 2020). The number of layers

available after each criterion was presented therein (Table 2). iv) As a last step, outliers were removed from the data, i.e., the mean intensive parameters in the layers were discarded when outside the following boundaries: 20 sr ≤ LR@355 ≤ 150 sr, 20 sr ≤ LR@532 ≤ 150 sr, -1 ≤ EAE ≤ 3, -1 ≤ BAE@355/532 ≤ 3, -1 ≤ BAE@532/1064 ≤ 3, 0 ≤ PDR@355 ≤ 0.3, and 0 ≤ PDR@355 ≤ 0.3. BAE represents the backscatter Ångström exponent and PDR represents the linear particle depolarization ratio (see Appendix A). As mentioned in Part 1, there was a low number of IPs removed (outliers) based on the above

predefined ranges (3.7%). In general, the number of the optical properties analysed is lower than the number of layers due to the following reasons: a) profiles of some optical properties are not available, b) some profiles do not cover the entire altitude range, c) the mean values are calculated only if 90% of the data are available while the SNR ≥ 2. Thus, we analysed a number of 795 layers for which we identified at least one intensive parameter.

The mean, median, minimum and maximum values of the intensive parameters for all of the stations providing at least one

parameter (except Sofia station) are shown in Table 1. The number of available values for each variable is shown as well (# lines).

In the current study, the smoke is considered fresh if LR@355 > LR@532 and EAE > 1.4 (Nicolae et al., 2013). Conversely, the smoke is considered aged when LR@532 > LR@355 and EAE < 1.4. LR denotes the lidar ratio and EAE the extinction Ångström exponent. These findings by Nicolae et al. (2013), based on lidar measurements were confirmed by measurements

with an aerosol mass spectrometer which allowed to estimate the degree of oxidation in BB aerosol. The colour ratio CR (spectral ratio) is the ratio of an optical parameter or an intensive parameter at two wavelengths (Appendix A). Here we refer to the colour ratio for the following intensive parameters: LR, BAE and PDR. We investigate the values of the CRs and we expect that they can be associated with fresh or aged smoke (short versus long distance smoke transport) and further be a fingerprint of the smoke as compared with other types of aerosols.

An event represents a series of BB measurements over a specific period of time. Thus, an event and a period are interchangeable. A measurement represents the data acquisition for a specific time, where the time is the average over 1 h. A measurement can contain one or more layers over the vertical profiles.



## 3 Biomass burning events originating in North America

In total, 24 events (periods of measurements) of smoke originating in North America were identified, for which at least one intensive parameter was retrieved. The events occurred in 2009 and 2011 and during 2012–2017. Eight events represented measurements of smoke coming solely from North America ('pure North America'), while the others represented 'mixed' smoke (mixture of North American and local smoke, i.e., fires were found along backtrajectory both in North America and locally). "Local smoke" refers to smoke originating in European locations, in general. In a few cases, the smoke came from North Africa or Middle East. The number of fires as well as the number of their detections (a fire can be detected more than once) are quantified.

### 3.1. Statistics on smoke originating from North America

There were 78 measurements over the 24 periods for which the backtrajectories of the layers met the criteria for North America origin, i.e., one period in 2009 and 2011, two periods in 2012 and 2014, three periods in 2015, and five periods in 2013, 2016 and 2017. All these periods lasted one day, except for the period 8–10 July 2013. From those, 8 events (51 measurements) represent pure North America smoke and 16 events (27 measurements) represent mixed smoke.

Figure 1 shows the intensive parameters and layers altitudes measured during the LRT of smoke from North America (smoke originating in North America is shown in black and mixed smoke in blue). At a first glance, there is no evidence of a systematic difference between the two categories. The mean (line), minimum and maximum (shaded areas) values from literature are displayed in red (for the variables displayed by '*') and green (for the variables displayed by 'o') corresponding to the references presented in Table 2. Compared to the values found in the rather limited existing literature for smoke originating in North America and measured over Europe (only tropospheric measurements were considered), we noted several IP values (especially for BAE@355/532) that fall outside of the range reported. The large value for the mixed smoke EAE may be due to the contribution of the local, fresh smoke. At a closer look, the large 'pure N America' EAE value, recorded on 4 July 2013 in Thessaloniki in a layer at ~ 3.6 km altitude, corresponds to air masses reaching ~ 9 – 11 km over the fires in North America. It is possible that the fires did not reach that altitude and thus the measurements for that layer may come from other sources. On the other hand, biomass burning particles can be found even in the lower stratosphere (e.g., Hu et al., 2019). The smallest EAE value (negative) may be due to dust contamination for a measurement performed in Granada on 19 August 2013 at 20:45 UTC, when fires in Portugal and North America were found along the backtrajectory. We also observe that mean PDR values are in general smaller if compared to the mean over the values reported in literature. However, still within the extreme values for smoke originating in North America (see Fig. 1 and Table 2). The minimum value reported for PDR@355 was 0.01±0.001 and for PDR@532 0.023±0.003 (Janicka et al., 2019). An EAE extreme value of -0.3 was reported by Haarig et al. (2018) but for the stratospheric smoke.

Overall, based on the mean values, we observed for North America fire particles advected over Europe a moderate absorption at 355 nm and a high absorption at 532 nm ($CR_{LR} > 1$), with low depolarization at both wavelengths, relatively large EAE





(apart from 2 isolated cases) - indication of small particles, slightly larger BAE@355/532 than BAE@532/1064. $CR_{LR}$ and EAE suggest the presence of aged particles, while BAE shows more backscatter for smaller wavelengths indicating small particles.

**4 Analyses of biomass burning over geographical regions**

A few studies are available on BB where the analysis is performed by examining different European regions by ground-based lidars. Baars et al. (2019) discuss the stratospheric smoke originating from Canada measured over different regions in Europe. The authors studied the event over six months and information about the change in optical depth, extinction, depolarization as well as the estimation of the mass concentration and ice nucleating particles are provided. Sicard et al. (2019) discuss the LRT of smoke plumes as measured over the Iberian Peninsula by means of ground/space, passive/active remote sensing and modelling. Observations and dispersion modelling altogether suggest that the particle depolarization properties are enhanced during their vertical transport from the mid to the upper troposphere. Ortiz-Amezcua et al. (2017) discuss the microphysical properties of the LRT smoke from North America over three lidar stations in Europe (Granada, Leipzig and Warsaw). It was shown that the layers accounted for ~ 40%, 30% and 70% of the total AOD for the three stations, respectively. Colour ratio of lidar ratios was around 2 while EAE was < 1.

The locations of the fires which produced the smoke layers detected by the stations located in South-East, South-West, North-East and Central Europe, and their histogram are shown in Figs. S2. For a straightforward comparison, we reproduce the figure for the South-East region from Part 1. Note that the grid size is 1°×1° (longitude×latitude).

For the South-East region, we distinguished a number of 321 fires located in North America (4.3%) and 7127 elsewhere, most of them located in East Europe. Most of the fires were located over [20°E 30°E] and [37°N 46°N], which corresponds to the Balkan region, covering parts of Romania, Bulgaria, North Macedonia, and Greece. Most of the measurements were taken at Bucharest, Athens and Thessaloniki stations.

For the South-West region, we identified a number of 197 fires in North America (8.7%) and 2066 elsewhere, most of the latter being located in the Iberian Peninsula and North Africa. Most of the fires occurred in the region [0° 10°W] x [35°N 43°N] that corresponds mostly to the Iberian Peninsula. Other fires were located over [0° 20°E] x [30°N 40°N], corresponding to North Africa (mostly North Algeria) and Sicily in South Italy. Most of the measurements were taken at Granada station.

For the Central Europe region, we have found 1420 fires originating in North America (86.9 %) and 214 elsewhere, most of the latter located in East Europe. Most of the fires occurred over [80°W 75°W] x [51°N 53°N] region, which corresponds to North America (East Canada). Most of the measurements in Cabauw station were performed over the LRT of smoke from North America, contributing to the histogram peak indicating North American locations. The stations of Hohenpeißenberg and Leipzig contained a ~24 % and ~79 % LRT of smoke from North America, but their number of measurements is much smaller than that of the Cabauw.



For the North-East region, 2761 of fires identified were located in North America (30.7 %), and 6228 elsewhere, most of the latter being located in East Europe (Ukraine and West Russia). Two peaks of the histogram indicate locations from North America ([75°W 78°W] x [51°N 53°N] and 95°W x [57°N 59°N]), which correspond to measurements taken at Belsk and Warsaw stations. Most of the measurements in East Europe belong to the grids delimited by [20°E 40°E] x [46°N 53°N] (mostly in Ukraine).

In summary, the main fire sources are located in: East Europe (especially Ukraine and West Russia), South Europe (Iberian Peninsula, Italy, Balkan region) and North America. Wildfires in the West Russian regions and Ukraine occur each year from March to October. Events of small particles transport, in the boundary layer, from these regions to the North-West Europe (Belarus, Poland, Germany, Nordic countries and European Arctic) are regularly recorded (Lund Myhre et al., 2007). Such transport of biomass burning aerosol can be extremely fast and affect relative humidity within the boundary layer (Stachlewska et al., 2017b). Transport of such particles to Arctic regions is contributing to arctic haze by significantly alternating the arctic aerosol properties (Stachlewska and Ritter, 2010), and thus contributing to Arctic warming.

The histogram of the backtrajectories (Figs. S3) revealed some preferential air circulation patterns for three of the regions (Central, South-West and North-East), with one common pattern being the circulation over the Atlantic. For the South-West region, we identified a vortex type circulation over North Africa as the main air pathway. For the North-East region we observed other patterns as well: a circulation from Iberian Peninsula, a circulation from East Europe (Caspian Sea), and a circulation over North Europe (Scandinavian Peninsula and West Russia).

### 4.1. Intensive parameters by geographical regions

A statistical investigation of the intensive parameters was performed, based on the continental fire source origin. As mentioned in Part 1, the following continental source origins were considered: Europe (EU), Africa (AF), Asia (AS), North America (NA), and combinations of two or more (e.g., EUAF=EU+AF, etc). The statistical analysis was performed over all of the available cases, despite their low number. Thus, we label the series with less than five cases as low statistics, and thus the results are just indicative and it is not safe to draw any conclusion. We reproduce here the results for South-East for a straightforward comparison with the other three regions. To thoroughly assess the smoke type, the scatter plots of EAE and $CR_{LR}$ are used.

Assuming that the aerosol size for the smoke layers is not significantly changing (e.g., Papanikolaou et al., 2020, for 532 nm), the LR is an indication for the absorption capacity of the particles and thus, the following description of the absorption (based on the values of the LR) can be further used: low absorption for LR < 40 sr, medium absorption for 40 sr < LR < 60 sr, high absorption for 60 sr < LR < 80 sr and very high absorption for LR > 80 sr.

### 4.1.1. South-East region

The mean values of the IPs in South-East region are shown in Fig. 2. One can observe the largest number for EU source region for all IPs. As for the other source regions we have low statistics. The mean LR values are between 40-60 sr (medium


absorption) except the larger values at LR@355 for the AS source region and LR@532 for the NA source region (low statistics). LR@355 is larger than LR@532 for EUAF and EUNA indicates fresh smoke and can be explained by a larger contribution to the mixed smoke from EU region. The mean value for EAE for the EU source region is 1.4 which suggests a mixture of fresh and aged smoke. Based on low statistics, we observe an increase of EAE from EUAF source region to EUAS

5    and EUNA source regions which corresponds to aged, fresh/aged and fresh smoke. Thus, the contribution of the local smoke in the mixed smoke is larger for EUNA. Similar values for LR and EAE for EU and EUAS suggest the larger contribution to the smoke from EU region. BAE values are similar for all source regions while BAE@355/532 > BAE@532/1064 for EU, AF and EUNA source regions. The PDR values are similar (except EU source regions, the others are just indicative). The scatter plots between various IPs do not show specific trends. Those plots are available in Supplement (Fig. S4.1).

### 4.1.2. North-East region

The mean values for the IPs are shown in Fig. 3. The majority of the events were recorded for the EU source region. LR@532 is slightly larger than LR@355 for EU source region, around 75sr. Based on low statistics, the two LR values for other source regions tend to be different. EAE ~1.4 is obtained for the EU source region which suggests a mixture of fresh and aged smoke. It would be worth investigating further in the future to see if the decrease of EAE from EUAF towards EUAS and EUNA

holds, based on different local contribution to the mixed smoke. BAE values are similar, except for AS (low statistics), where are larger. BAE@355/532 is larger than BAE@532/1064 for all source regions, which denotes more backscatter at 355 nm. The similarity between NA and EUNA source regions suggests a major contribution from NA to the EUNA mixture. PDR@532 is larger than PDR@355, except for EUNA source region. As expected, the scatter plots between various IPs show a linear regression between the two PDR and between the two BAE (Figs. S4.2). Large values of LR for EU suggest more

absorption, if compared to the South-East region.

### 4.1.3. South-West region

The mean values for the IPs for South West region are shown in Fig. 4. LR values are similar, slightly larger for EUNA. LR@355 is larger than LR@532 for EU and AF (low statistics) source regions. A few values available for EAE indicate aged smoke. Small EAE values for EUAF and EUNA (low statistics) indicate large contribution from Africa and North America,

respectively. Large, similar values are observed for BAE for NA and EUNA source regions, suggesting that the contribution to EUNA is more from North America. Smaller values are observed for EUAF source region. BAE@355/532 is larger than BAE@532/1064 for all but AF source region. Based on scatter plots (Figs. S4.3), we observe a direct proportionality between the two BAEs (observed also for North-East region).

### 4.1.4. Central region

The mean values for the IPs in the Central region are shown in Fig. 5. The number of IPs is very small and thus, the observations are not statistically significant, although there seems to be a tendency towards low absorption (LR ~40 sr). EAE for EU source



region suggests fresh smoke, while the two values for NA indicate aged smoke. The large value for EAE for EUAF region needs to be investigated (e.g., if the contribution of the local fires is predominant). BAE values are similar for the two source regions (EU and NA). Due to the small number of values, the scatter plots are not statistically significant (Figs. S4.4). Average values were performed for EU source region. For the scatter plot between the two BAEs, a mean value from NA source region

was also available.

### 4.2 Statistical analysis over all regions

We perform the analysis based on the mean IP values as a function of continental source region. We consider analysing the scatter plots between the different CRs and EAE, where, for each scatter plot, the mean values correspond to the same measurements. Still, different scatter plots can refer to slightly different sets of measurements.

As a general statement, we consider that the cases where we have only one or two measurements are not statistically significant, a good confidence is considered when at least five measurements are available. Therefore, the results discussed below should be carefully treated in such situations.

### 4.2.1 Observations based on scatter plots

The general observations based on the scatter plots between CR or EAE are shown in Fig. 6. Each point on the graph represents

the average for one measurement region (South-East, South-West, Central and North-East Europe) and one continental source (EU, AF, AS, NA, EUAF, EUAS, EUNA). All available data are averaged. We added for comparison the mean values (red circles, Fig. 6) found in literature (Table S1, Part 1). For the North-East region, the PDRs provided by the Warsaw station allowed a complete comparison. For a better visualization of the mean CR (Fig. 6) and the corresponding IPs, in Fig. 7 are shown the CR and IP values versus continental source regions (i.e., the panels a–f of Fig. 7 corresponds to the a–f scatter plots

of Fig. 6). The right-hand side axis shows the number of available measurements for the scatter plots. The black line represents the number of five cases.

For increasing $CR_{PDR}$ we found an increase of the EAE (Fig. 6b), while the $CR_{LR}$ decreases (Fig. 6c). Based on panel b), we observe that except the case with low EAE (<0.5) and $CR_{PDR}$<1 which indicates aged particles and larger depolarization at 355 nm, the depolarization at 532nm can be higher for either fresh or aged smoke. The dataset is not statistically significant, but

increased number of samples in future studies is expected to reveal the statistical significance of this correlation. A slight decrease of the $CR_{PDR}$ with smoke travel time was observed (see lower values for EUNA, NA, AS and EUAS), while the $CR_{BAE}$ maintained similar values for all the source regions. An increase of EAE versus decreasing $CR_{LR}$ (Fig. 6d), evident especially for the North-East region (Fig. 7d), was reported also by Samaras et al. (2015) and Janicka et al. (2019).

No clear relationship between $CR_{BAE}$ and $CR_{LR}$ (Fig. 6e), $CR_{BAE}$ and EAE (Fig. 6f) and $CR_{BAE}$ and $CR_{PDR}$ (Fig. 6a) was found.

Veselovskii et al. (2015) showed that the relationship between EAE and BAE is not straightforward, pointing out that while EAE depends mainly on the particle size, BAE depends on both particle size and complex refractive index. The relationship between BAE and EAE was analysed from the relative humidity (RH) perspective by Su et al. (2008) and Wang et al. (2019).



They showed that the relationship of BAE and EAE depends on the RH values and, thus, one can find correlated and anti-correlated behaviours. However, the RH influence is out of the scope of this study. Still, we made use of RH values for an individual case shown below.

As seen in Fig. 7, different features are observed for different measurement regions. Based on the EAE–$CR_{BAE}$ scatter plot
(Fig. 6f), Fig. 7f indicates for the source regions EUAF, EUAS and EUNA the following. For the North-East region, EAE decreases (from EUAF towards EUAS and EUNA), while both BAE increase, but $CR_{BAE}$ is similar. For the South-West region, EAE and both BAE increase from EUAF to EUNA source regions, while $CR_{BAE}$ is similar. For the South-East region, EAE increases, while no signature is found for BAE and $CR_{BAE}$. The Central region provides data for the EU and NA source regions. Here, EAE, both BAE and $CR_{BAE}$ decrease from EU to NA source regions. Considering the findings of Veselovskii et al.
(2015), we conclude that, for the Central region, the fine particle mode is predominant for the EU source region (as compared with the NA source region), result which is expected. For the South-East region we find a larger amount of fine particles for EUNA source region as compared to EUAS and EUAF source regions. This implies a large contribution of the EU source region to the mixture. For the North-East measurement region, we also find an increase for BAE@355/532, while based on the LR and $CR_{LR}$ values, the absorption at 532 nm increases from EUAF towards EUAS and EUNA.

**4.2.2 Continental source regions**

Based on the results shown in Figs. 6 and 7, one can assess how distinct are the characteristics of the smoke coming from various continental source regions as observed in different geographical regions. The mean values are shown in Table 3 for each of the d)–f) scatter plots presented in Fig. 6. The smoke type (fresh versus aged) is assessed based on the values of the $CR_{LR}$ and EAE. Information on the smoke absorption and depolarization ratio (where available) is provided. There is no clear
relationship between $CR_{BAE}$ and EAE or $CR_{LR}$ (see Fig. 6d–f), while a slight decrease of EAE with increasing $CR_{LR}$ is captured. The highlighted values in Table 3 show the occurrences with low statistics and thus, more corroborated results are needed in the future to draw unambiguous conclusions.

Except for one isolated case, we obtained positive values for BAE (and $CR_{BAE}$), which indicates more backscattering towards smaller wavelengths, and low depolarization ratios (all PDR < 0.1). Except for two extremes (-1.6 and 3.2), all $CR_{BAE}$ values
range between 0.18 and 1.6. The $CR_{PDR}$ (available for the North-East region only) has the largest value for the EUAF source region, followed by EU and EUAS. The lowest $CR_{PDR}$ and EAE values were found for the EUNA source region, characterized also by the highest $CR_{LR}$ (aged smoke; less depolarizing and more absorbing at 532 nm). The high EAE values of the smoke mixtures are likely due to the large EU contribution (EAE value for the North-East region with EUAF origin is 1.46, for the South-East region with EUAS origin is 1.5, and for the South-East region with EUNA origin is 1.9). Table 4 summarizes the
key observations over BB layers according to its source and measurement region.

The main features are the following. In the South-East region generally aged smoke with the EU source was measured. For the other source regions (low statistics), the smoke was labelled as aged for NA and EUAF regions, a mixture of fresh and aged smoke from the EUAS source region and fresh smoke from EUNA source region. For the mixed source regions (EUAF,



EUAS and EUNA), the contribution of the local (EU) determine if the smoke type was fresh or aged. The EU source region provided medium absorbing particles. The other values are indicative (low statistics). The smoke from EUAS and EUNA show medium absorption (as for EU) suggesting the influence of EU smoke contribution. However, the particles size is slightly smaller, based on EAE values. The smoke from AF and NA indicates lower absorption at 355nm. High absorption is observed

for EUAF source region and it does not resemble either EU or AF regions.

In the South-West region (low statistics except EUAF), aged and highly absorbing smoke particles from all source regions were measured. For the EU and AF source regions (which have similar EAE and LR values), we assume aged smoke, based on the high RH (where $CR_{LR} < 1$ and EAE ~ 1).

In the Central region (low statistics), aged smoke from EU and NA source regions was measured, displaying a low absorption

for the EU source region and higher backscatter values at 1064 nm for the NA source region. More corroborative measurements are needed to draw solid conclusions.

The North-East region displayed mixed fresh and aged smoke from the EU source region (highly absorbing). For the other regions, the few values are only indicative. For EUAF source region, we measured fresh smoke (probably due to EU contributions) with less absorption at 532nm. The smoke from EUAS and EUNA was labelled as aged, showing very high

absorption at 532nm.

 Higher/lower depolarization at 355/532 nm was observed for the LRT of smoke from North America (as for the North-East region). Based on a single continental source, in all regions, aged smoke was measured, except the North-East with a mixture of fresh and aged smoke from the EU source region. Based on two continental sources (mixtures), the regions measure either aged, fresh or mixed (aged and fresh) smoke, depending on the lower or higher contribution of the local source.

**5 Summary and conclusions**

The present study shows results based on the biomass burning events as measured by EARLINET over the 2008–2017 period, according to a methodology described in Part 1 (Adam et al., 2020). The aerosol layers were labelled as smoke based on a combined analysis of Hysplit backtrajectories and the FIRMS fire locations. The smoke was further labelled as 'mixed' if multiple fire sources contributed to the smoke measurement. For the smoke originating in North America, the smoke was

labelled as 'pure North America' or 'mixed' (with contribution from fires in Europe). We demonstrated that in most of the cases the smoke was mixed and the quantification (based on number of fires and detections) of the contributing fires to the mixture explains the wide range of values obtained for the intensive parameters.

The statistics over all LRT events from North America revealed no significant difference between the measurements where the smoke was originating solely from North America and the measurements with mixed smoke (having origin in both North

America and local). This suggests that the contribution of the local smoke is not significant. Based on the LR values, a moderate absorption at 355 nm (46 sr) and a high absorption at 532 nm (66 sr) were observed. The mean $CR_{LR}$ and EAE suggest aged



smoke, while the PDR values indicate a low depolarization and the BAE reveal higher backscatter values at smaller wavelengths.

The statistical analysis of the smoke properties (fresh versus aged and absorption capability) in four European regions (Central, North-East, South-West and South-East Europe) separating the smoke events into continental source regions (European, North American, African, Asian or a mixture of European with each of the remaining), based on trajectory analysis revealed the following. The smoke detected in Central Europe (Cabauw, Leipzig, and Hohenpeißenberg) was mostly brought form North America (87% of fires). In North-East Europe (Belsk, Minsk, Warsaw), the smoke was advected mostly from Eastern Europe (Ukraine and Russia) but there was a significant contribution (31%) of smoke from North America. In South-West Europe (Barcelona, Evora, Granada) smoke originated mainly in Iberian Peninsula and North Africa, (while 9% was originating in North America. In South-East Europe (Athens, Bucharest, Potenza, Sofia, Thessaloniki) the origin of the smoke was mostly local (only 3% of smoke from North America.

For each region, the IPs were analysed based on their continental source origin.

The analysis of the scatter plots revealed correlated with the increase of smoke travel time (corresponding to aging), $CR_{LR}$ and $CR_{BAE}$ increase while EAE and $CR_{PDR}$ decrease. These tendences, associated with the smoke characteristics, can be further used when analysing various types of aerosols and thus helping identifying the smoke among other aerosol types. The variability of the mean values / standard deviation (STD) was large in general and, thus, the individual values for different source regions overlap. Based on data from Warsaw (North-East region), the depolarization at 532 nm decreases for LRT smoke from North America (while $CR_{PDR} < 1$).

Smoke was found to be aged in all measurement regions (except North-East) if there is no mixture among different fires. On the contrary, when the origin of the smoke has two continental sources, either aged, fresh or a mixture of aged and fresh smoke can be measured, depending on the smaller or higher contribution of the European (local) sources. Thus, in the South-East measurement region, fresh smoke from the EUNA source region and a mixture of fresh and aged smoke originating from the EUAS was measured. In the North-East, region fresh smoke originating from EUAF was measured.

For the South-West region with European or African source regions we obtained a $CR_{LR}$ of 0.8 and an EAE of 1. We assumed that the smoke measured was aged based on the high RH (in agreement with Veselovskii et al., 2020). The lowest absorption was determined for the Central region (LRs < 36 sr). The South-West region displayed a highly absorbing smoke (61 sr < LR@355 < 79 sr and 64 < LR@532 < 91 sr). The South-East region displayed smoke with a medium/high absorption at 532 nm (50–72 sr) and a low/medium absorption at 355 nm (31–48 sr). The smoke measured in the North-East region has a medium to very high absorption at 532 nm (57–91 sr) and a medium to high absorption at 355 nm (46–78 sr).

The quite diverse absorption was determined for the different measurement's regions, even for smoke from the same continental source region, which may be related, among others, with different RH conditions (e.g., Veselovskii et al, 2020). In line with previous studies, we showed that BAE and further $CR_{BAE}$ do not show specific values based on sources and no trends, and thus, they cannot be used to identify the smoke type. In order to easily quantify the smoke type, LR ($CR_{LR}$) and EAE are essential. The aerosol typing algorithm developed by Papagiannopoulos et al. (2018) based on 3 backscatter and 2



extinction input provides one category for smoke. NATALI (Nicolae et al, 2018) distinguishes between smoke, continental smoke and mixed smoke if depolarization data are available additionally. Based on the implementation of ACTRIS Research Infrastructure in the next few years, the presented methodology will be applied on a larger dataset (more automatic lidar systems expected) providing a more complete (3 backscatter + 2 extinction + 1-3 depolarization) datasets with enhanced

quality control procedures.

The present methodology shows new approaches for smoke characterization (smoke type along with information on absorption and depolarization in the context of different continental sources) and provides valuable information for various scientific communities (modelling, satellite).. The analysis reported in the paper shows the potentialities of the used approach for identifying specific features of smoke particles in different geographical regions and for long range transported cases. The

obtained results will be corroborated by the increasing number of aerosol profiling data coming into the EARLINET database thanks to the implementation of ACTRIS (Aerosol Clouds Trace Gases Research Infrastructure). This process is currently reducing the time delay in data provision, improving the quality of data products and increasing also the number of multi wavelength lidar systems over Europe. This extension of the observations will allow in the near future to increase the statistics of the result obtained with the approach here presented.

For further investigations we envisage a more detailed analysis on grouping the sources' locations using cluster analysis, where a larger number of clusters should be chosen to identify more homogeneous regions with similar vegetation type. Thus, a more accurate correlation between the source type and the measurements is envisaged. Moreover, the smoke time travel will be integrated. The challenge that remains is the quantification of the contribution of different fires in the mixed smoke (besides their number and detections). EartCARE future mission could provide this kind of information about the smoke coverage and

transport (https://earth.esa.int/eogateway/missions/earthcare, last access 19 July 2021).



# Appendix A

**Table A1. List of acronyms**

| Nomenclature | Definition |
|---|---|
| ACTRIS | Aerosol Cloud and Trace Gases Research Infrastructure |
| a.g.l. | Above ground level |
| a.s.l. | Above sea level |
| "atz", "brc", "cog", "ino", "cbw", "evo", "gra", "lei", "mas", "hpb", "pot", "sof", "the", "waw" | Athens, Barcelona, Belsk, Bucharest, Cabauw, Evora, Granada, Leipzig, Minsk, Hohenpeißenberg, Potenza, Sofia, Thessaloniki and Warsaw<br><br>(lidar stations considered in this study) |
| BAE | Backscatter Ångström exponent. $BAE@355/532=-\log(\beta p355/\beta p532)/\log(355/532)$, $BAE@355/532=-\log(\beta p532/\beta p1064)/\log(532/1064)$ |
| BB | Biomass burning |
| $\beta_P$ | Particle backscatter coefficient [1/m/sr] |
| CR(s) | Colour ratio(s). $CR_{LR}=LR@532/LR@355$, $CR_{BAE}=BAE@532/1064/BAE@355/532$, $CR_{PDR}=PDR@532/PDR@355$ |
| EAE | Extinction Ångström exponent. $EAE@355/532=-\log(\kappa p355/\kappa p532)/\log(355/532)$ |
| EARLINET | European Aerosol Research Lidar Network |
| EU, AF, NA, AS | Europe, Africa, North America, Asia continental source regions |
| EUAF, EUNA, EUAS | Europe + Africa, Europe + North America, Europe + Asia continental source regions |
| FIRMS | Fire Information for Resource Management System |
| FRP | Fire radiative power |
| GDAS | Global Data Assimilation System |
| HYSPLIT | Hybrid Single-Particle Lagrangian Integrated Trajectory model |
| IP(s) | Intensive parameter(s) |
| $\kappa_P$ | Particle extinction coefficient [1/m] |
| LR | Lidar ratio [sr]. $LR@355=\kappa p355/\beta p355$, $LR@532=\kappa p532/\beta p532$ |
| LRT | Long range transport |
| MODIS | Moderate Resolution Imaging Spectroradiometer |
| PDR | Linear particle depolarization ratio |
| QC | Quality control |
| SCC | Single Calculus Chain. See D'Amico et al., EARLINET Single Calculus Chain – overview on methodology and strategy, Atmos. Meas. Tech., 8, 4891–4916, doi:10.5194/amt-8-4891-2015, 2015. |
| SE, SW, CE and NE | Southeast, Southwest, Central and Northeast Europe (geographical measurement regions) |
| SNR | Signal to noise ratio. It is defined as the ratio of the signal to its uncertainty. |
| STD | Standard deviation |





Appendix B

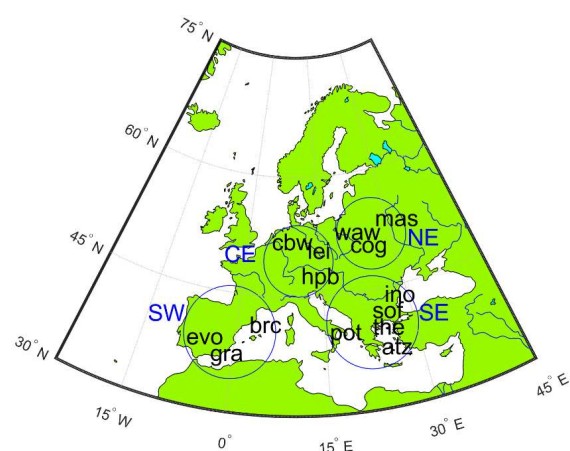

**Figure B1. The geographical location of the 14 stations providing data for Forest Fire category in EARLINET database over 2008-2017 period. The stations are located in Athens ("atz"), Barcelona ("brc"), Belsk ("cog"), Bucharest ("ino"), Cabauw ("cbw"), Evora ("evo"), Granada ("gra"), Leipzig ("lei"), Minsk ("mas"), Hohenpeissenberg ("hpb"), Potenza ("pot"), Sofia ("sof"), Thessaloniki ("the") and Warsaw ("waw"). The blue circles show the four European geographical regions: South-East (SE), South-West (SW), North-East (NE) and Central (CE).**

**Table B1. Geographical coordinates for the lidar stations ([https://www.earlinet.org/index.php?id=105](https://www.earlinet.org/index.php?id=105), last access 19 August 2021)**

| Coordinates<br>Station ("code") | Latitude | Longitude | Altitude [m] | link |
|---|---|---|---|---|
| Athens ("atz") | 37.9600 N | 23.7800 E | 212 | http://www.physics.ntua.gr/~papayannis/ |
| Barcelona ("brc") | 41.3930 N | 2.1200 E | 115 | http://www.tsc.upc.edu/rslab/ |
| Belsk ("cog") | 51.8300 N | 20.7800 E | 180 | http://www.igf.edu.pl/ |
| Bucharest ("ino") | 44.3480 N | 26.0290 E | 93 | http://www.inoe.ro/en/ |
| Cabauw ("cbw") | 51.9700 N | 4.9300 E | 0 | http://projects.knmi.nl/earlinet/ |
| Evora ("evo") | 38.5678 N | 7.9115 W | 293 | http://www.icterra.pt/g1/ |
| Granada ("gra") | 37.1640 N | 3.6050 W | 680 | http://www.iista.es/ |
| Leipzig ("lei") | 51.3500 N | 12.4330 E | 125 | http://www.tropos.de/en/ |
| Minsk ("mas") | 53.9170 N | 27.6050 E | 200 | http://ifan.basnet.by/ |


| Hohenpeissenberg ("hpb") | 47.8019 N | 11.0119 E | 974 | http://www.dwd.de/EN/research/observing_atmosphere/composition_atmosphere/hohenpeissenberg/start_mohp_node.html |
|---|---|---|---|---|
| Potenza ("pot") | 40.6000 N | 15.7200 E | 760 | https://www.imaa.cnr.it/ |
| Sofia ("sof") | 42.6500 N | 23.3800 E | 550 | http://www.ie-bas.org/ie_Eng.htm |
| Thessaloniki ("the") | 40.6300 N | 22.9500 E | 50 | http://lap.physics.auth.gr/ |
| Warsaw ("waw") | 52.2100 N | 20.9800 E | 112 | https://www.igf.fuw.edu.pl/en/instruments/laboratorium-pomiarow-zdalnych-e91845-7230/ |

*Author contributions*. MA developed the methodology, analysed results and wrote the paper. All authors, except MA, contributed by conducting measurements, ensuring data quality, and performing data evaluation and data provision to the EARLINET Data Base. ISS, LM, NP, JABA and MS contributed with revisions of the paper. All authors read the paper and agreed with its content.

*Competing interests*. The authors declare that they have no conflict of interest.

*Special issue statement*. This article is part of the special issue "EARLINET aerosol profiling: contributions to atmospheric and climate research". It is not associated with any conference.

Acknowledgements:

*We acknowledge the use of data and imagery from LANCE FIRMS operated by the NASA/GSFC/Earth Science Data and Information System (ESDIS) with funding provided by NASA/HQ. The authors acknowledge the NOAA Air Resources Laboratory (ARL) for the provision of the HYSPLIT transport and dispersion model and/or the READY website (http://www.ready.noaa.gov) used in this publication. The authors acknowledge the EARLINET-ACTRIS community for provision of the aerosol lidar profiles used in this study, in particular those who performed measurements, evaluated lidar data and provided profiles to the Forest Fire category in the EARLINET-ACTRIS database. We acknowledge Wojciech Kumala, Krzysztof Markowicz, and Rafal Fortuna (University of Warsaw) for technical support at the ACTRIS site in Warsaw, and Cristi Radu, Dragos Ene and Alexandru Dandocsi (INOE 2000) for technical support at the ACTRIS site in Magurele.*

*Funding: The research leading to these results has received funding from the European Union Seventh Framework Programme (FP7/2007-2013) under grant agreement n° 262254, as well as the H2020 ACTRIS-2 grant n° 654109, ACTRIS PPP grant n° 739530, ACTRIS IMP grant n° 871115. It was also supported with following national funding: the Romanian National contracts 18N/08.02.2019, PN-III-P2-2.1-PED-2019-1816 and 19PFE/17.10.2018 as well as with the European Space Agency (ESA-ESTEC) funding: The Technical assistance for Polish Radar and Lidar Mobile Observation System*



(POLIMOS 4000119961/16/NL/FF/mg). The work is co-funded by the European Union through the European Regional Development Fund, included in the COMPETE 2020 (Operational Program Competitiveness and Internationalization) through the ICT project (UIDB/04683/2020) with the reference POCI-01-0145- FEDER-007690 and also through TOMAQAPA (PTDC/CTAMET/ 29678/2017). The research was partially funded by the European Regional Development

Fund through the Competitiveness Operational Programme 2014-2020, POC-A.1-A.1.1.1- F- 2015, project Research Centre for Environment and Earth Observation CEO-Terra, SMIS code 108109, contract No. 152/2016. Juan Antonio Bravo-Aranda received funding from the Marie Sklodowska-Curie Action Cofund 2016 EU project – Athenea3i under grant agreement no. 754446.

Data access: *The FIRMS data used in the study is available upon request from https://firms.modaps.eosdis.nasa.gov/ (last access: 13 July 2021). The data files used in this study as well as the output of the data analysis is available upon request (contact mail: mariana.adam@inoe.ro).*

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

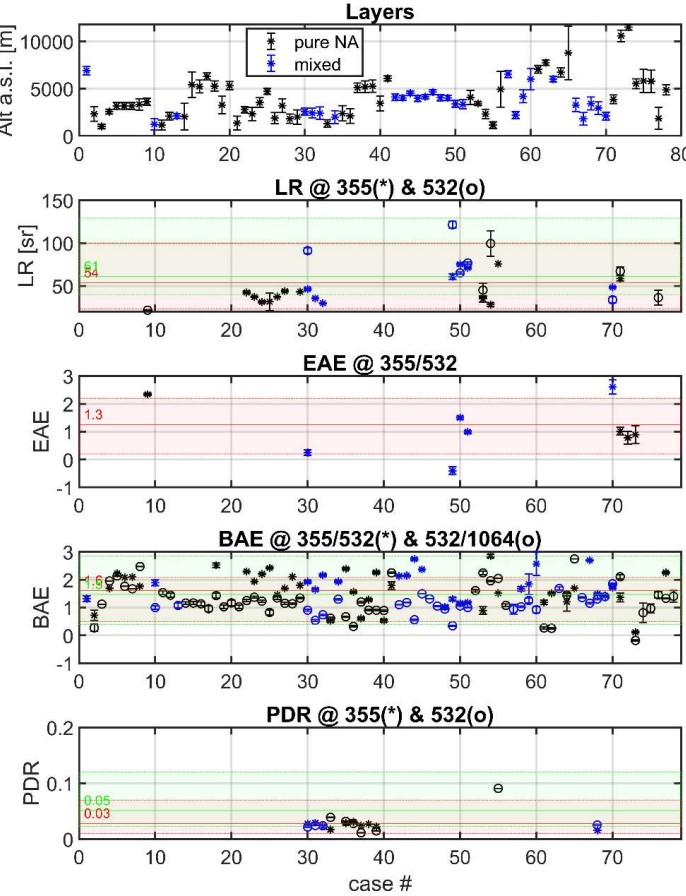

**Figure 1.** All of the 78 measurements recorded during LRT smoke from North America. Measurements are divided into North America origin ('pure NA' - black) and 'mixed' (North America and local - blue) origin. There are 27 measurements of mixed smoke. Along with the intensive parameters, the layers mean altitude and thickness (marked as error bar) are shown in the upper plot. Mean values from literature are shown with red (for * values) and green (for o values) lines while the shaded areas delineate the minimum and maximum values. 'pure NA' stands for 'pure North America'. Symbols (* and o) are shown in panels title.

3.4

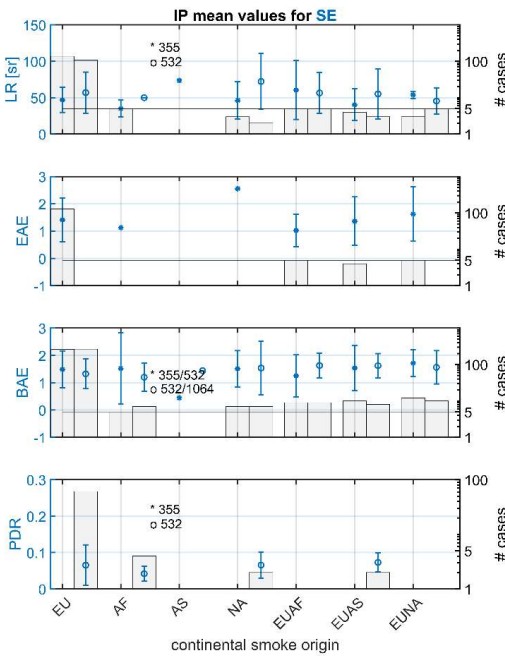

**Figure 2. The mean and standard deviation of the intensive parameters for the South-East (SE) region. The right axis shows the number of available values for each IP. The horizontal line represents 5 cases. The number of cases below this threshold are considered low statistics.**



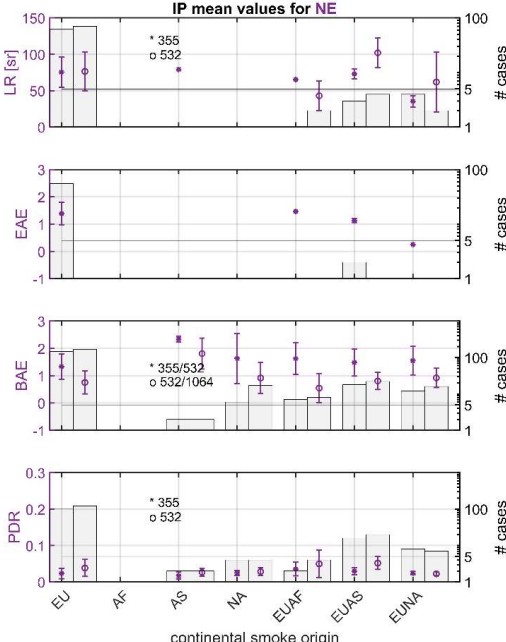

**Figure 3. The mean and standard deviation of the intensive parameters for the North-East (NE) region. The right axis shows the number of available values for each IP.**





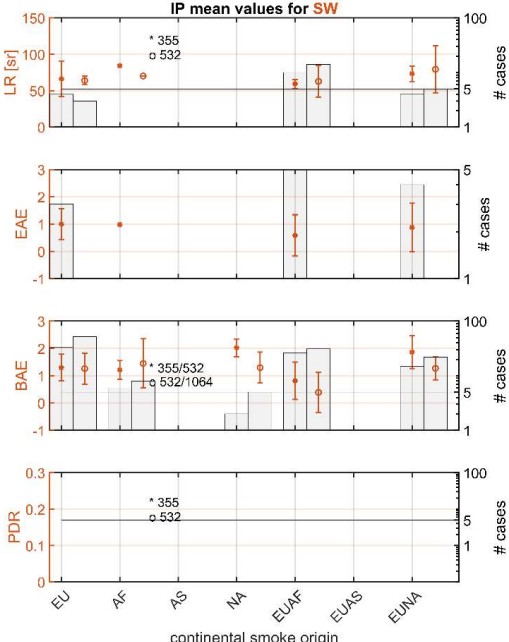

**Figure 4. The mean and standard deviation of the intensive parameters for the South-West (SW) region. The right axis shows the number of available values for each IP.**





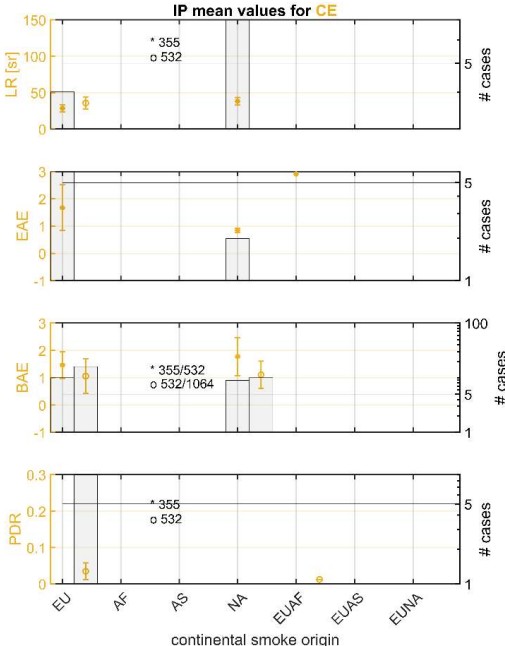

**Figure 5.** The mean and standard deviation of the intensive parameters for the Central Europe (CE) region. The right axis shows the number of available values for each IP.



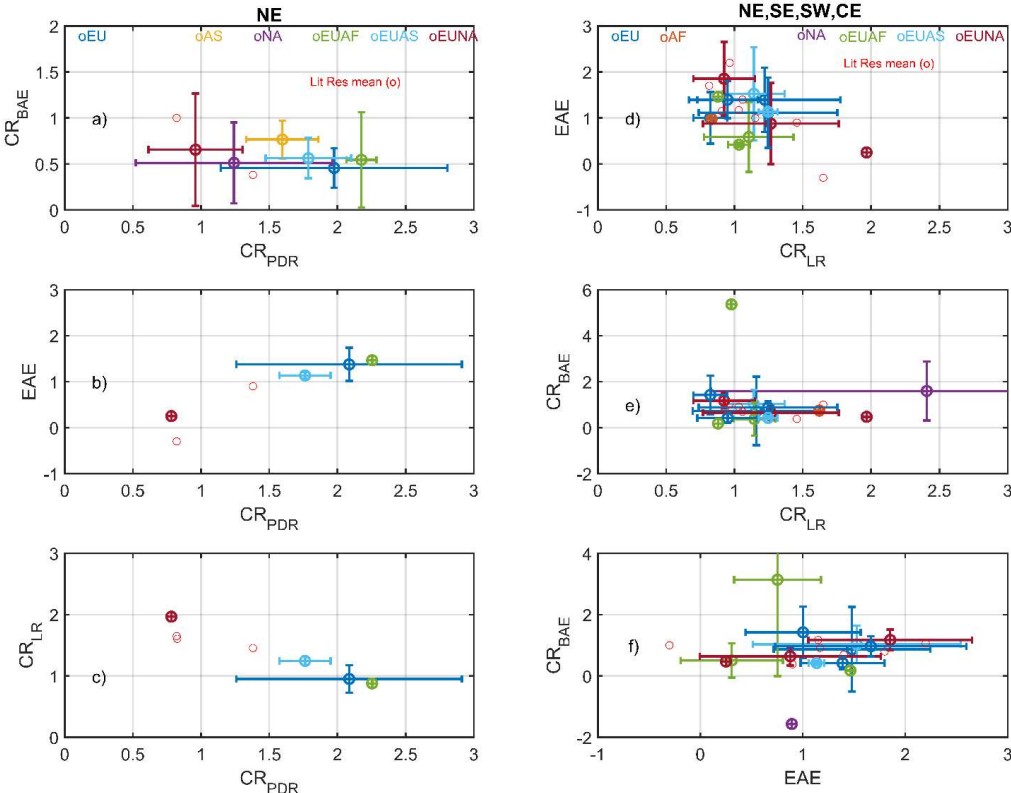

**Figure 6.** Scatter plots between (a) CR$_{BAE}$ and CR$_{PDR}$, (b) EAE and CR$_{PDR}$, (c) CR$_{LR}$ and CR$_{PDR}$, (d) EAE and CR$_{LR}$, (e) CR$_{BAE}$ and CR$_{LR}$, and (f) CR$_{BAE}$ and EAE. The mean values found in literature are shown in red circles. a)-c) plots are obtained only for the North-East (NE) region where two PDR values are available (Warsaw).





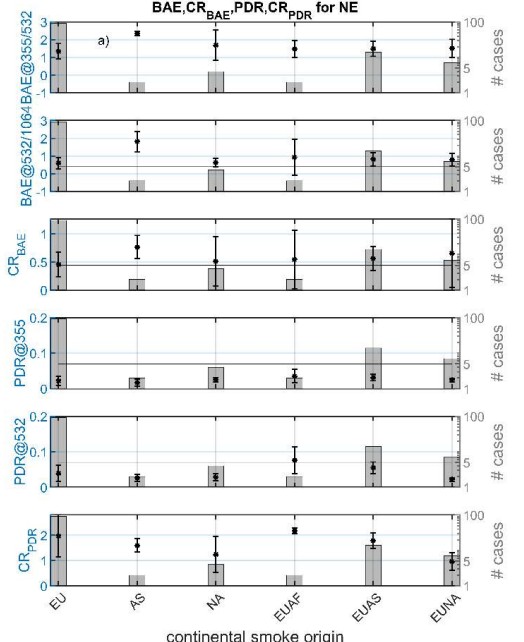

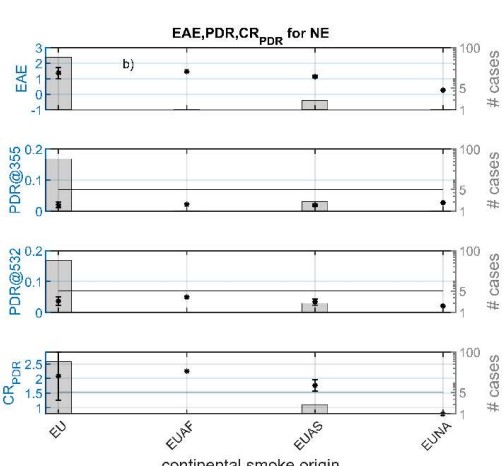





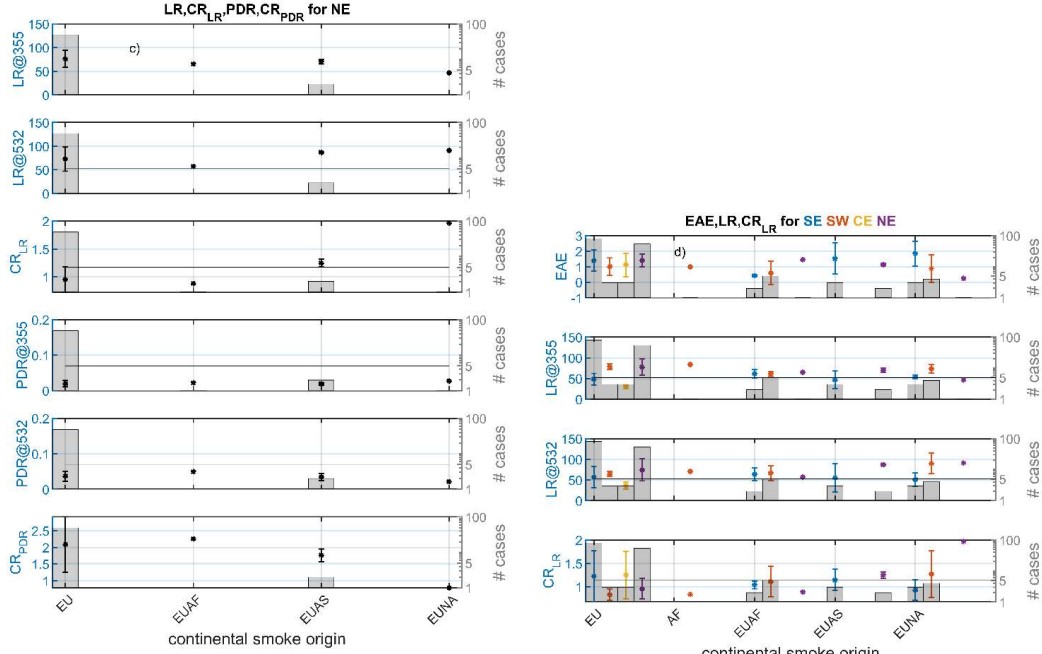





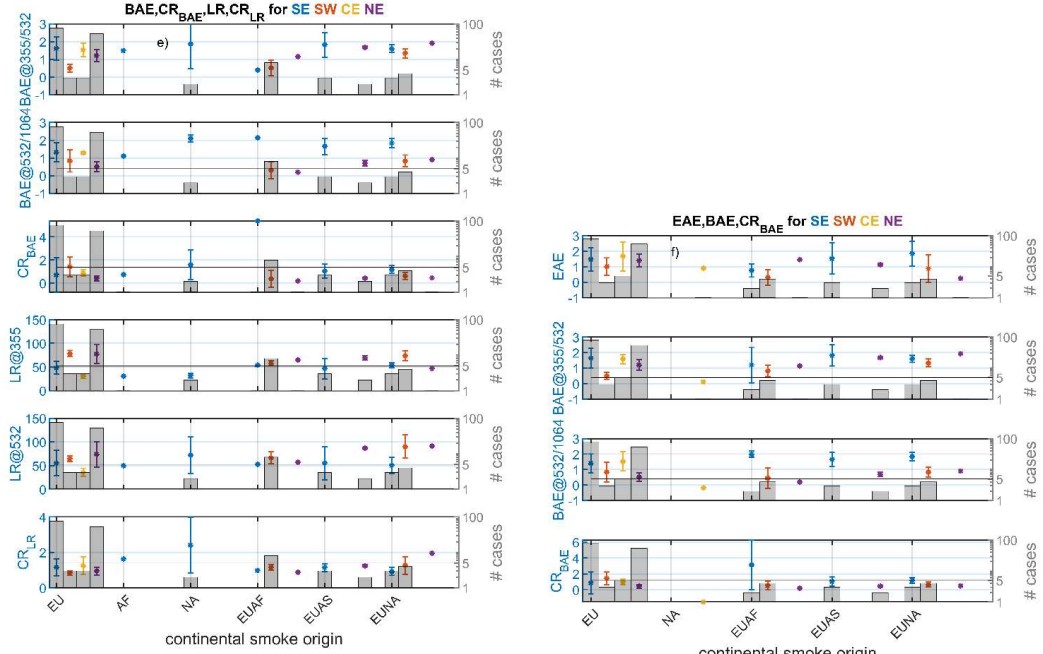

**Figure 7.** Intensive parameters and corresponding colour ratio (CR) versus smoke source origin. Plots a)–f) correspond to plots a)–f) in Fig. 6.





Atmospheric Chemistry and Physics Discussions — Open Access

**Table 1. Main features (mean, median, minimum, maximum values and associated uncertainties) of the intensive parameters.**

**¹ 355/532; ² 532/1064; ³ number of total values for a specific parameter. Mean values are highlighted. See stations' acronyms in Appendix A.**

| | | atz | brc | cog | ino | cbw | evo | gra | lei | mas | hpb | pot | the | waw |
|---|---|---|---|---|---|---|---|---|---|---|---|---|---|---|
| LR355 | #³ | 83 | 0 | 0 | 52 | 9 | 0 | 19 | 0 | 0 | 2 | 3 | 20 | 71 |
| | Mean | 47±2 | | | 51±2 | 35±3 | | 65±2 | | | 33±1 | 48±1 | 36±0.4 | 73±2 |
| | Med | 42±3 | | | 52±1 | 37±1 | | 61±1 | | | 33±1 | 51±1 | 32±0.5 | 74±1 |
| | Min | 20±1 | | | 25±1 | 24±1 | | 31±1 | | | 32±1 | 43±1 | 21±1 | 30±1 |
| | Max | 126±6 | | | 90±11 | 44±1 | | 86±7 | | | 33±1 | 51±1 | 65±1 | 128±2 |
| LR532 | #³ | 54 | 0 | 0 | 34 | 1 | 1 | 22 | 0 | 0 | 2 | 2 | 34 | 78 |
| | Mean | 57±5 | | | 57±1 | 46±2 | 36±9 | 68±3 | | | 31±1 | 55±3 | 58±1 | 77±2 |
| | Med | 50±1 | | | 53±1 | 46±2 | 36±9 | 66±1 | | | 31±1 | 55±1 | 50±1 | 72±1 |
| | Min | 21±3 | | | 29±1 | 46±2 | 36±9 | 40±4 | | | 30±1 | 27±2 | 20±1 | 29±1 |
| | Max | 142±8 | | | 115±1 | 46±2 | 36±9 | 121±4 | | | 33±1 | 82±3 | 133±3 | 146±2 |
| EAE¹ | #³ | 64 | 0 | 0 | 32 | 2 | 0 | 13 | 0 | 0 | 7 | 2 | 46 | 59 |
| | Mean | 1.4 ±0.2 | | | 1.3 ±0.02 | 1.3 ±0.3 | | 0.8 ±0.1 | | | 1.7 ±0.3 | 1.4 ±0.1 | 1.6 ±0.05 | 1.4 ±0.05 |
| | Med | 1.4 ±0.5 | | | 1.2 ±0.1 | 1.3 ±0.4 | | 1 ±0.04 | | | 1.7 ±0.1 | 1.4 ±0.2 | 1.8 ±0.1 | 1.3 ±0.02 |
| | Min | -0.8 ±0.1 | | | -0.6 ±0.01 | 0.3 ±0.1 | | -0.4 ±0.1 | | | 0.8 ±0.2 | 0.6 ±0.1 | -0.9 ±0.1 | 0.3 ±0.1 |
| | Max | 2.6 ±0.3 | | | 2.6 ±0.01 | 2.3 ±0.4 | | 1.7 ±0.3 | | | 2.9 ±0.2 | 2.3 ±0.1 | 2.7 ±0.2 | 2.8 ±0.2 |
| BAE¹ | #³ | 113 | 5 | 5 | 113 | 14 | 2 | 77 | 0 | 37 | 5 | 4 | 78 | 150 |
| | Mean | 1.6 ±0.1 | 1.4 ±0.2 | 2.1 ±0.1 | 1.4 ±0.03 | 1.8 ±0.01 | 2 ±0.03 | 1.2 ±0.04 | | 1.3 ±0.2 | 1.2 ±0.02 | 1 ±0.05 | 1.5 ±0.1 | 1.4 ±0.05 |
| | Med | 1.6 ±0.1 | 1.2 ±0.3 | 2.2 ±0.05 | 1.5 ±0.006 | 1.8 ±0.1 | 2 ±0.1 | 1.3 ±0.001 | | 1.2 ±0.2 | 1.4 ±0.01 | 0.8 ±0.1 | 1.5 ±0.1 | 1.3 ±0.15 |
| | Min | -1 ±0.01 | 1.2 ±0.03 | 1.5 ±0.1 | -0.1 ±0.0004 | 0.5 ±0.01 | 1.7 ±0.03 | -0.2 ±0.02 | | 0.4 ±0.1 | 0.1 ±0.03 | 0.7 ±0.03 | -0.8 ±0.03 | 0.4 ±0.1 |
| | Max | 2.9 ±0.01 | 1.8 ±0.02 | 2.5 ±0.1 | 2.8 ±0.1 | 2.4 ±0.03 | 2.3 ±0.04 | 2.9 ±0.03 | | 2.6 ±0.1 | 1.8 ±0.02 | 1.7 ±0.1 | 3 ±0.01 | 2.8 ±0.1 |
| BAE² | #³ | 110 | 14 | 20 | 119 | 14 | 8 | 98 | 6 | 35 | 6 | 3 | 76 | 176 |
| | Mean | 1.4 ±0.04 | 1.2 ±0.04 | 1.2 ±0.1 | 1.4 ±0.02 | 1.1 ±0.05 | 1.3 ±0.1 | 1 ±0.01 | 0.9 ±0.1 | 0.8 ±0.1 | 1.2 ±0.02 | 1.3 ±0.01 | 1.3 ±0.03 | 0.7 ±0.03 |
| | Med | 1.3 ±0.1 | 1.1 ±0.4 | 1.2 ±0.2 | 1.3 ±0.01 | 1.2 ±0.1 | 1.3 ±0.2 | 1.1 ±0.04 | 1.1 ±0.1 | 0.7 ±0.03 | 1.3 ±0.1 | 1.3 ±0.01 | 1.2 ±0.2 | 0.7 ±0.11 |
| | Min | 0.8 ±0.02 | 0.7 ±0.01 | 1 ±0.1 | 0.1 ±0.003 | 0.6 ±0.04 | 0.8 ±0.3 | -0.7 ±0.04 | -0.6 ±0.1 | -0.9 ±0.1 | -0.2 ±0.02 | 1.3 ±0.02 | 0.1 ±0.002 | -0.2 ±0.03 |
| | Max | 2.8 ±0.05 | 1.8 ±0.01 | 1.5 ±0.1 | 2.8 ±0.01 | 1.4 ±0.03 | 1.6 ±0.01 | 2.9 ±0.01 | 1.6 ±0.05 | 1.9 ±0.1 | 2.6 ±0.04 | 1.3 ±0.01 | 3 ±0.01 | 2.2 ±0.03 |
| PDR355 | #³ | 0 | 0 | 0 | 0 | 0 | 0 | 0 | 0 | 0 | 0 | 0 | 0 | 132 |
| | Mean | | | | | | | | | | | | | 0.024 ±0.0002 |
| | Med | | | | | | | | | | | | | 0.022 ±0.01 |
| | Min | | | | | | | | | | | | | 0.002 |





|  |  |  |  |  |  |  |  |  |  |  |  |  |  |
|---|---|---|---|---|---|---|---|---|---|---|---|---|---|
|  |  |  |  |  |  |  |  |  |  |  |  |  | ±0.00001 |
|  | Max |  |  |  |  |  |  |  |  |  |  |  |  | 0.086 ±0.0004 |
| PDR532 | #³ | 0 | 0 | 0 | 64 | 0 | 0 | 0 | 0 | 0 | 10 | 5 | 0 | 160 |
|  | Mean |  |  |  | 6.6 ±0.3 |  |  |  |  |  | 3.3 ±0.1 | 4.5 ±0.1 |  | 0.039 ±0.0004 |
|  | Med |  |  |  | 4.9 ±0.6 |  |  |  |  |  | 2.4 ±1.3 | 4.9 ±0.1 |  | 0.034 ±0.012 |
|  | Min |  |  |  | 0.04 ±0.001 |  |  |  |  |  | 1.2 ±0.03 | 2.3 ±0.1 |  | 0.006 ±0.0001 |
|  | Max |  |  |  | 27.5 ±1.5 |  |  |  |  |  | 8.1 ±0.1 | 6.1 ±0.1 |  | 0.151 ±0.001 |





**Table 2. Long range transport events with fire sources in North America. Mean and STD of the intensive parameters. The number of cases available is given in parenthesis (# cases). Pure NA stands for smoke originating in North America solely.**

| Intensive parameter | LR 355 mean ± STD [sr] | LR 532 mean ± STD [sr] | EAE 355/532 mean ± STD | BAE 355/532 mean ± STD | BAE 532/1064 mean ± STD | PDR 355 mean ± STD [%] | PDR 532 mean ± STD [%] |
|---|---|---|---|---|---|---|---|
| All (# cases) | 46 ± 16 (18) | 66 ± 32 (10) | 1.1 ± 0.9 (9) | 1.7 ± 0.6 (53) | 1.2 ± 0.5 (74) | 0.024 ± 0.005 (10) | 0.031 ± 0.022 (10) |
| Pure NA (# cases) | 42 ± 14 (11) | 54 ± 30 (5) | 1.3 ± 0.7 (4) | 1.7 ± 0.7 (32) | 1.3 ± 0.6 (48) | 0.025 ± 0.005 (6) | 0.036 ± 0.029 (6) |
| Mixed (# cases) | 52 ± 17 (7) | 78 ± 32 (5) | 1 ± 1.2 (5) | 1.8 ± 0.5 (21) | 1.1 ± 0.3 (26) | 0.024 ± 0.006 (4) | 0.024 ± 0.002 (4) |
| Lit res* (# cases) min, max | 54 ± 10 (9) 23, 100 | 61 ± 3 (12) 40, 129 | 1.3 ± 0.4 (6) 0.2, 2.2 | 1.6 ± 0.1 (6) 0.5, 2.1 | 1.5 ± 0.03 (8) 0.4, 2.85 | 0.03 ± 0.03 (3) 0.01, 0.07 | 0.05 ± 0.006 (6) 0.02, 0.12 |

*Literature research: according to references 4,13,15,19,20,23,26,31,36,46** from Table S4, Part 1 (Adam et al., 2020). Current mean values are the averages over the values reported (see text). Minimum and maximum values are shown as well.

** 4.    Ancellet, G., Pelon, J., Totems, J., Chazette, P., Bazureau, A., Sicard, M., Di Iorio, T., Dulac, F., and Mallet, M.: Long-range transport and mixing of aerosol sources during the 2013 North American biomass burning episode: analysis of multiple lidar observations in the western Mediterranean basin, Atmos. Chem. Phys., 16, 4725–4742, doi:10.5194/acp-16-4725-2016, 2016.

13.    Groß, S., Esselborn, M., Weinzierl, B., Wirth, M., Fix, A, and Petzold, A.: Aerosol classification by airborne high spectral resolution lidar Observations, Atmos. Chem. Phys., 13, 2487–2505, doi:10.5194/acp-13-2487-2013, 2013.

15.    Haarig, M., Ansmann, A., Baars, H., Jimenez, C., Veselovskii, I., Engelmann, R., and Althausen, D.: Depolarization and lidar ratios at 355, 532, and 1064 nm and microphysical properties of aged tropospheric and stratospheric Canadian wildfire smoke, Atmos. Chem. Phys., 18, 11847-11861, https://doi.org/10.5194/acp-18-11847-2018, 2018.

19.    Janicka, L., Böckmann, C., Wang, D., Stachlewska, I. S.: Lidar derived fine scale resolution properties of tropospheric aerosol mixtures, ILRC29, S2-122, Hefei, China, 2019.

20.    Janicka, L., Stachlewska, I. S., Veselovskii, I., Baars, H.: Temporal variations in optical and microphysical properties of mineral dust and biomass burning aerosol derived from daytime Raman lidar observations over Warsaw, Poland, Atmos. Environ., 169, 162-174, http://dx.doi.org/10.1016/j.atmosenv.2017.09.022, 2017.

23.    Mattis, I., Müller, D., Ansmann, A., Wandinger, U., Preißler, J., Seifert, P., and Tesche, M.: Ten years of multiwavelength Raman lidar observations of free-tropospheric aerosol layers over central Europe: Geometrical properties and annual cycle, J. Geophys. Res., 113, D20202, doi:10.1029/2007JD009636, 2008.

26.    Müller, D., Kolgotin, A., Mattis, I., Petzold, A., and Stohl, A.: Vertical profiles of microphysical particle properties derived from inversion with two-dimensional regularization of multiwavelength Raman lidar data: experiment, Appl. Opt., 50, 2069-2079, 2011.

31.    Ortiz-Amezcua, P., J. L. Guerrero-Rascado, M. J. Granados-Muñoz, J. A. Benavent-Oltra, C. Böckmann4, S. 30 Samaras, I. S. Stachlewska, Ł. Janicka, H. Baars, S. Bohlmann, and L. Alados-Arboledas, Microphysical characterization of long-range





transported biomass burning particles from North America at three EARLINET stations, Atmos. Chem. Phys., 17, 5931–5946, doi:10.5194/acp-17-5931-2017, 2017.

36.      Preißler, J., F. Wagner, J. L. Guerrero-Rascado, and A. M. Silva, Two years of free-tropospheric aerosol layers observed over Portugal by lidar, J. GEOPHYS. RES., 118, 3676–3686, doi:10.1002/jgrd.50350, 2013.

46.      Wandinger, U., D. Müller, C. Böckmann, D. Althausen, V. Matthias, J. Bo¨senberg, V. Weiß, M. Fiebig, M. Wendisch, A. Stohl, and A. Ansmann, Optical and microphysical characterization of biomass burning and industrial pollution aerosols from multiwavelength lidar and aircraft measurements, J. Geophys. Res., 107, NO. D21, 8125, doi:10.1029/2000JD000202, 2002.





**Table 3. Mean values and their STD for IPs for each region (SE, SW, CE, NE) and each continental source region (EU, AF, NA, EUAF, EUAS, EUNA). The first column block refers to the scatter plot in Fig. 6d (EAE versus CR$_{LR}$), the middle column block refers to the scatter plot in Fig. 6e (CR$_{LR}$ versus CR$_{BAE}$) and the last column block refers to the scatter plot in Fig. 6f (EAE versus CR$_{BAE}$). n represents the number of events available for each scatter plot. The highlighted values correspond to low statistics.**

| SE | CR$_{LR}$ | LR532 | LR355 | EAE | n | CR$_{LR}$ | LR532 | LR355 | CR$_{BAE}$ | BAE2 | BAE1 | n | EAE | CR$_{BAE}$ | BAE2 | BAE1 | n |
|---|---|---|---|---|---|---|---|---|---|---|---|---|---|---|---|---|---|
| EU | 1.2±0.6 | 57±26 | 48±14 | 1.4±0.7 | 81 | 1.2±0.5 | 55±27 | 48±13 | 0.7±1.5 | 1.3±0.5 | 1.6±0.6 | 76 | 1.5±0.8 | 0.9±1.4 | 1.4±0.6 | 1.6±0.6 | 80 |
| AF | | | | | | 1.6±0 | 50±0 | 31±0 | 0.7±0 | 1.1±0 | 1.5±0 | 1 | | | | | |
| NA | | | | | | 2.4±1.6 | 72±38 | 32±5 | 1.6±1.3 | 2.1±0.2 | 1.9±1.4 | 2 | | | | | |
| EUAF | 1.0±0.1 | 64±15 | 61±10 | 0.4±0.1 | 2 | 1.0±0 | 53±0 | 54±0 | 5.4±0 | 2.1±0 | 0.4±0 | 1 | 0.8±0.4 | 3.1±3.1 | 2.0±0.2 | 1.2±1.1 | 2 |
| EUAS | 1.1±0.2 | 55±35 | 47±22 | 1.5±1 | 3 | 1.1±0.2 | 55±35 | 47±22 | 1.0±0.6 | 1.7±0.5 | 1.8±0.7 | 3 | 1.5±1 | 1.0±0.6 | 1.7±0.5 | 1.8±0.7 | 3 |
| EUNA | 0.9±0.2 | 50±17 | 54±5 | 1.9±0.8 | 3 | 0.9±0.2 | 50±17 | 54±5 | 1.2±0.3 | 1.8±0.3 | 1.6±0.2 | 3 | 1.9±0.8 | 1.2±0.3 | 1.8±0.3 | 1.6±0.2 | 3 |

| SW | CR$_{LR}$ | LR532 | LR355 | EAE | | CR$_{LR}$ | LR532 | LR355 | CR$_{BAE}$ | BAE2 | BAE1 | | EAE | CR$_{BAE}$ | BAE2 | BAE1 | |
|---|---|---|---|---|---|---|---|---|---|---|---|---|---|---|---|---|---|
| EU | 0.8±0.1 | 64±6 | 78±7 | 1.0±0.6 | 3 | 0.8±0.1 | 64±6 | 78±7 | 1.4±0.8 | 0.8±0.6 | 0.5±0.2 | 3 | 1.0±0.6 | 1.4±0.8 | 0.8±0.6 | 0.5±0.2 | 3 |
| AF | 0.8±0 | 70±0 | 84±0 | 1.0±0 | 1 | | | | | | | | | | | | |
| NA | | | | | | | | | | | | | | | | | |
| EUAF | 1.1±0.3 | 67±18 | 61±6 | 0.6±0.8 | 5 | 1.1±0.2 | 67±12 | 58±5 | 0.4±0.7 | 0.3±0.5 | 0.5±0.4 | 8 | 0.3±0.5 | 0.5±0.6 | 0.4±0.7 | 0.8±0.4 | 4 |
| EUAS | | | | | | | | | | | | | | | | | |
| EUNA | 1.3±0.5 | 90±25 | 73±11 | 0.9±0.9 | 4 | 1.3±0.5 | 90±25 | 73±11 | 0.6±0.3 | 0.8±0.3 | 1.3±0.3 | 4 | 0.9±0.9 | 0.6±0.3 | 0.8±0.3 | 1.3±0.3 | 4 |

| CE | CR$_{LR}$ | LR532 | LR355 | EAE | | CR$_{LR}$ | LR532 | LR355 | CR$_{BAE}$ | BAE2 | BAE1 | | EAE | CR$_{BAE}$ | BAE2 | BAE1 | |
|---|---|---|---|---|---|---|---|---|---|---|---|---|---|---|---|---|---|
| EU | 1.2±0.5 | 36±9 | 30±4 | 1.1±0.8 | 3 | 1.2±0.5 | 36±9 | 30±4 | 0.9±0.3 | 1.3±0.1 | 1.5±0.4 | 3 | 1.7±0.9 | 1.0±0.3 | 1.5±0.6 | 1.6±0.3 | 5 |
| AF | | | | | | | | | | | | | | | | | |
| NA | | | | | | | | | | | | | 0.9±0 | -1.6±0 | -0.2±0 | 0.1±0 | 1 |
| EUAF | | | | | | | | | | | | | | | | | |
| EUAS | | | | | | | | | | | | | | | | | |
| EUNA | | | | | | | | | | | | | | | | | |

| NE | CR$_{LR}$ | LR532 | LR355 | EAE | | CR$_{LR}$ | LR532 | LR355 | CR$_{BAE}$ | BAE2 | BAE1 | | EAE | CR$_{BAE}$ | BAE2 | BAE1 | |
|---|---|---|---|---|---|---|---|---|---|---|---|---|---|---|---|---|---|
| EU | 0.9±0.2 | 74±27 | 78±19 | 1.4±0.4 | 54 | 1.0±0.2 | 74±27 | 78±19 | 0.4±0.2 | 0.5±0.3 | 1.2±0.3 | 53 | 1.4±0.4 | 0.4±0.2 | 0.5±0.3 | 1.2±0.3 | 54 |
| AF | | | | | | | | | | | | | | | | | |
| NA | | | | | | | | | | | | | | | | | |
| EUAF | 0.9±0 | 57±0 | 65±0 | 1.5±0 | 1 | 0.9±0 | 57±0 | 65±0 | 0.2±0 | 0.2±0 | 1.1±0 | 1 | 1.5±0 | 0.2±0 | 0.2±0 | 1.1±0 | 1 |
| EUAS | 1.2±0.1 | 87±1 | 70±5 | 1.1±0.1 | 2 | 1.2±0.1 | 87±1 | 70±5 | 0.4±0.1 | 0.7±0.1 | 1.7±0.1 | 2 | 1.1±0.1 | 0.4±0.1 | 0.7±0.1 | 1.7±0.1 | 2 |
| EUNA | 2.0±0 | 91±0 | 46±0 | 0.3±0 | 1 | 2.0±0 | 91±0 | 46±0 | 0.5±0 | 0.9±0 | 1.9±0 | 1 | 0.3±0 | 0.5±0 | 0.9±0 | 1.9±0 | 1 |



**Table 4. Smoke characteristics based on $CR_{LR}$ and EAE for each measurement region (South-East - SE, South-West - SW, North-East - NE and Central - CE) and each source region (Europe - EU, Africa - AF, Asia - AS, North America - NA, Europe + Africa - EUAF, Europe + Asia - EUAS, Europe + North America - EUNA) based on scatter plots in Fig. 6. The highlighted values correspond to low statistics.**

| | $CR_{LR}$ | EAE | LR532 (sr)* | LR355 (sr)* | $CR_{BAE}$** | Comments | Smoke type based on $CR_{LR}$ and EAE*** | Absorption at 532nm and 355nm**** |
|---|---|---|---|---|---|---|---|---|
| **SE** | | | | | | | | |
| EU | 1.2 | 1.4 | 57 | 48 | 0.8 | 81 meas. $CR_{LR}$ and EAE | aged | Medium |
| AF | 1.6 | | 50 | 31 | 0.74 | 1 meas. $CR_{LR}$ and $CR_{BAE}$ | aged | Medium at 532nm Low at 355nm |
| NA | 2.4 | | 72 | 32 | 1.6 | 2 meas. $CR_{LR}$ based on $CR_{LR}$ versus $CR_{BAE}$ | aged | High at 532nm Low at 355nm |
| EUAF | 1 | 0.4 | 64 | 61 | 3.2 | 2 meas. $CR_{LR}$ and EAE | aged | High |
| EUAS | 1.1 | 1.5 | 55 | 47 | 1 | 3 meas. $CR_{LR}$ and EAE larger EU contribution | fresh/aged | Medium |
| EUNA | 0.9 | 1.9 | 51 | 54 | 1.2 | 3 meas. $CR_{LR}$ and EAE larger EU contribution | fresh | Medium |
| **SW** | | | | | | | | |
| EU | 0.8 | 1 | 64 | 78 | 1.4 | 3 meas. $CR_{LR}$ and EAE RH=68–70%. | aged | High |
| AF | 0.8 | 1 | 70 | 84 | | 1 meas. $CR_{LR}$ and EAE. RH=73%. | aged | High at 532nm Very high at 355nm |
| EUAF | 1.1 | 0.6 | 67 | 61 | 0.5 | 5 meas. $CR_{LR}$ and EAE | aged | High |
| EUNA | 1.3 | 0.9 | 90 | 73 | 0.6 | 4 meas. $CR_{LR}$ and EAE | aged | Very high at 532nm High at 355nm |
| **CE** | | | | | | | | |
| EU | 1.2 | 1.1 | 36 | 30 | 0.9 | 3 meas. $CR_{LR}$ and EAE | aged | Low |
| NA | | 0.9 | | | -1.6 (-0.2 / 0.1) | 1 meas. EAE and $CR_{BAE}$. More backscattering at 1064nm. | aged | |
| **NE** | | | | | | | | |
| EU | 1 | 1.4 | 74 | 78 | 0.4 | 54 meas. $CR_{LR}$ and EAE $CR_{PDR} > 1$ | fresh/aged | High |
| EUAF | 0.9 | 1.5 | 57 | 65 | 0.2 | 1 meas. $CR_{LR}$ and EAE larger EU contribution $CR_{PDR} > 1$ | fresh | Medium at 532nm High at 355nm |
| EUAS | 1.1 | 1.1 | 87 | 70 | 0.4 | 2 meas. $CR_{LR}$ and EAE $CR_{PDR} > 1$ | aged | Very high at 532nm High at 355nm |
| EUNA | 2 | 0.3 | 91 | 46 | 0.5 | 1 meas. $CR_{LR}$ and EAE $CR_{PDR} < 1$ | aged | Very high at 532nm Medium at 355nm |

\* corresponding to $CR_{LR}$; ** based on $CR_{BAE}$ versus $CR_{LR}$ and/or EAE; *** Based on scatter plot between EAE and $CR_{LR}$ where available (Fig. 6, upper right-hand side); **** LR is considered low for <40sr, medium for [40,60]sr, high for [60,80]sr, very high for > 80sr.