# Peer review of "Biomass burning events measured by lidars in EARLINET - Part 2: Optical properties investigation."

_Atmospheric Chemistry and Physics, 2021_

## Author Comment (AC1)

We would like to thank both reviewers for reading carefully our manuscript and for their constructive comments.

We respect the reviewers' decisions, whatever they are.

Please see below our comments (in blue) while in red we specify the text from the manuscript. The new adding is highlighted. Different citations are shown in green.

**RC 2**

Comment on acp-2021-759

Anonymous Referee #1

Referee comment on "Biomass burning events measured by lidars in EARLINET – Part 2: Optical properties investigation" by Mariana Adam et al., Atmos. Chem. Phys. Discuss., https://doi.org/10.5194/acp-2021-759-RC2, 2021

The manuscript deals with lidar-derived optical properties of fire smoke in the air over different parts of Europe. The observations cover fresh smoke events and events with aged smoke after long range transport (frequently from North America). The European Aerosol Research Lidar Network (EARLINET) data base is used for this study. Unfortunately, the amount of data is so low and even not of high quality (to my opinion) that clear messages and convincing conclusions cannot be drawn. I clearly see that the first author did her best, however, the available data set does not allow to write a sound scientific 'story' because numerous and high quality results are not available. Mariana Adam can just present some kind of a final report (of a not just very successful, big field campaign lasting over many, many years), rather than an article with well extracted results and findings. This is impossible with the used (poor) data set.

At the end, after careful reading and realizing how poor the available data set is, I cannot recommend publication. If this report would be published in its present form, most of the readers would conclude: Better not to use any EARLINET data. The amount is low and the quality is quite questionable, and in full contrast to what aerosol scientists expected when they are familiar with, e.g., the AERONET data base.

Regarding the amount of data, we would like to specify the following. As mentioned in Part I, the Earlinet measurements on biomass burning (Forest Fire category) were perform based on voluntary basis. Thus, each station contributed as much as it could, based on their availability to perform out of schedule measurements. On one hand, the number of such events over Central Europe is much smaller compared with South Europe. On the other hand, even for some of the Southern stations, there are limitations in providing smoke measurements due to the fact that many of them are contaminated with Saharan dust (and thus, not reported in this category). For the available records, some data were dismissed during quality control. Table 1 shows the final number of available data for the 13 stations providing at least one intensive parameter. Most of the measurements were provided by Athens, Bucharest, Thessaloniki and Warsaw, followed by Granada. Further, when we consider the (measurements) geographical regions, we obtain a good record for SE and NE measurement regions. The methodology considered the differentiation of the sources based on their continental origin (Europe, Africa, N America, Asia or a mixture of two). A good statistical coverage was obtained for sources located in Europe for measurements taken in SE and NE regions. Moreover, even for SE and NE measurement regions the number of IPs for smoke sources located over other continents or a mixture of two is small (Figs 2 and 3). Recall that for LRT pollution, the measured signal is not always strong and thus, suitable to

retrieve the extinction coefficient and further LR or EAE. Further, when performing scatter plots, the number of coincident measurements where two IPs are retrieved, can be small (Tables 3 and 4). Consequently, we mentioned that these results are indicative for the time being. We believe that future measurements with smoke sources located on other continents besides Europe will strengthen the trends shown in Figure 6. Unfortunately, this takes some time to record suitable measurements.

**RC 1**

This is a follow-on study from Adam et al (2020), using the results of that analysis to examine the properties and origins of smoke layers detected by the EARLINET network. The enormous amount of data collected by EARLINET and the careful analysis presented in the first paper provides the opportunity for some meaningful statistics, which are very much needed as we become ever more aware of forest fires in our warming world. However, the paper requires major revision before being suitable for publication.

• The greatest weakness in the presentation of the paper lies in the Introduction, where we learn almost nothing about the results of previous studies of smoke transport – simply providing a long list of references is not sufficient to set the context. What have other authors found? What is the gap in their results that this paper is going to fill? A more thorough Introduction needs to prepare the reader for the methodology and approach that this paper uses. As it is, the paper just presents statistics with no context. A cursory 'literature survey' pops up in the results section (Section 4 page 6 lines 5-14), but this is not the right place nor it is anywhere near comprehensive enough.

Section 4 page 6 lines 5-14 was moved to Introduction. We revised the whole Introduction. The following paragraphs were included:

The most common synergy in studying the BB smoke employs the lidar and photometry measurements.

The studies regarding the fresh smoke revealed the following. Balis et al (2003) estimate the optical properties, revealing a  $CR_{LR} < 1$  and  $EAE \sim 1.8$  (denoting small particles) while single scattering albedo (SSA) was 0.9 at 355 nm (denoting weak absorption) and AOD of the smoke layer was 1.15 (85% of the total AOD). Alados-Arboledas (2011) determined the smoke optical and microphysical properties, revealing a pronounced accumulation mode (particles smaller than 500 nm) with the mean effective radius of 0.2 µm, SSA of 0.76-0.9 (ranging from 355 nm to 1064 nm) while the smoke layer represented 50 % of the total AOD. For this study,  $CR_{LR} \sim 1$  while EAE was ~1.3. Sicard et al. (2012) studied the Saharan dust and BB events and computed the direct radiative forcing, showing that the Saharan dust had a positive forcing while the smoke had a negative forcing in top of the atmosphere. On the other hand, the Saharan dust showed a stronger absorption. Nicolae et al. (2013) studied both fresh and aged smoke using lidar and mass spectroscopy and they showed that in general, the fresh smoke has  $CR_{LR} < 1$  and EAE > 1.4 while it is more absorbing than the aged smoke.

The smoke optical properties were investigated during various meteorological conditions. Thus, Stachlewska et al. (2017a) study a heat wave over Poland and they showed an increase of the PBL height and total AOD among others. Osborne et al. (2019) investigated the Saharan dust and smoke over UK during an ex-hurricane. The authors classified the Saharan dust and BB smoke function of PDR and modelled air-masses trajectories. A multilayer structure was observed with either Saharan dust or smoke origin and they estimated a higher load (mass concentration) for the smoke.

Regarding the aged smoke, the following were observed. Wandinger et al. (2002) studied the optical and the microphysical properties using lidar and insitu measurements. An effective radius of 0.25  $\mu$ m

was reported while SSA was 0.78 - 0.83 at 532 nm (absorbing smoke) and layer AOD was 0.1. Mattis et al. (2003) reported EAE over 0.6 - 1.2 and  $CR_{LR} > 1$ , small depolarization. It was observed that LR@532 decreased with time but it was still larger than LR@355. Muller et al. (2005) studied Canadian and Siberian smoke over Germany. For Siberian smoke, it was found CRLR > 1, EAE over 0 -1.3 range and effective radius over  $0.24 - 0.41 \mu m$ . The smoke from Canada showed an effective radius  $\leq 0.2 \ \mu$ m, while the CRLR < 1 and EAE over 1.8 - 2.1. The authors assumed that anthropogenic sources were responsible for these values. SSA in all cases varied between 0.9 and 0.98 (low absorption). It was shown that EAE decreases with increasing effective radius. Ancellet at al. (2016) analysed pure BB, weakly dusty BB, BB - dust mixture and Saharan dust based on PDR, employing also modelling (FLEXPART) and Caliop measurements of North American smoke transported over Mediterranean region. Thus, the authors measured PDR of < 5 % for pure BB, between 5 and 10 % for weakly dusty BB,  $10 \pm 2$  % for BB - dust mixture and  $30 \pm 2$  % for Saharan dust. Haarig et al. (2018) studies the spectral dependence of PDR and microphysical properties of aged tropospheric and stratospheric smoke where the PDR and LR at three wavelengths were for the first time reported. The tropospheric smoke showed spherical particles (PDR < 3 % at all wavelengths) while the stratospheric smoke particles showed PDR of 22 %, 18 % and 4 % at 355 nm, 532 nm and 1064 nm respectively. The LR were similar for tropospheric and stratospheric smoke (similar scattering and absorption properties): 40 - 45 sr, 65 - 80 sr and 80 - 95 sr at 355 nm, 532 nm and 1064 nm respectively. The estimated effective radius was 0.32 µm in stratosphere and 0.17 µm in troposphere while the estimated mass concentration was 5.5  $\mu$ g m-3 in troposphere and 40  $\mu$ g m-3 in stratosphere. SSA in stratosphere was estimated to be 0.74, 0.8 and 0.83 at 355 nm, 532 nm and 1064 nm respectively. Stachlewska et al. (2018) studied the optical properties of the fresh and aged smoke during a highpressure system. The aged smoke showed  $CR_{LR} > 1$  and EAE < 1.49 and PDR smaller than for fresh smoke. During smoke intrusion, the PBL height increased as well as the AOD and EAE while the ground insitu showed an increase of PM10 and PM2.5. Vaughan et al. (2018) presented a Canadian smoke transported over UK which persisted several days over North-Western Europe during an atmospheric block. The smoke layers became optically thinner but the LR and PDR showed little change. The highest OD (~ 0.15) was observed for layers above 7 km. PDR was in the range of 4-6% below 7 km and close to 20 % above 7 km. LR at 355 nm was in the range of 35 - 65 sr. It was inferred that the nature of the smoke for different layers was different. Hu et al. (2019) studied the Canadian smoke transported in the stratosphere over France. PDR at the three wavelengths (355 nm, 532 nm and 1064 nm) was found around 20 %, 18 - 19 % and 4 - 5 % (similar with Haarig et al, 2018). Effective radius was estimated to 0.33  $\mu$ m while SSA (0.8 – 0.9) showed absorbing smoke particles. The direct radiative forcing showed the reduction of the radiation arriving at the surface. The highest OD of the stratospheric layer was  $\sim 0.2$  at 532 nm and CRLR > 1.

A few studies are available on BB where the analysis is performed by examining different European regions with ground-based lidars. Baars et al. (2019) discuss stratospheric smoke originating from Canada measured over different regions in Europe. The authors studied the event over six months and information about the change in optical depth, extinction, depolarization as well as the estimation of the mass concentration and ice nucleating particles are provided. Sicard et al. (2019) discuss the LRT of smoke plumes as measured over the Iberian Peninsula by means of ground/space, passive/active remote sensing and modelling. Observations and dispersion modelling altogether suggest that the particle depolarization properties are enhanced during their vertical transport from the mid to the upper troposphere. Ortiz-Amezcua et al. (2017) discuss the microphysical properties of the LRT smoke from North America over three lidar stations in Europe (Granada, Leipzig and Warsaw). It was shown that the layers accounted for ~ 40%, 30% and 70% of the total AOD for the three stations, respectively. Colour ratio of lidar ratios was around 2 while EAE was < 1.

The concluding remarks from the literature review are the following. CRLR and EAE are indicative for the delimitation of the fresh versus aged smoke. The PDR of the tropospheric smoke is smaller than

PDR for the stratospheric smoke. The effective radius is around  $0.2 - 0.4 \mu m$  while the smoke can show weaker or stronger absorption (based on SSA calculations) for both smoke types. Note that the effective radius, SSA or the radiative forcing can be retrieved or calculated if at least 3 particle backscatter coefficients and 2 particle extinctions coefficients are determined.

The current study aimed to find specific features for LRT smoke advected from N America (1) and to find specific features for smoke originating from different continents and recorded in different geographical regions in Europe (2). According to the methodology shown in Part I, in many cases, the recorded smoke is a mixture of different smoke (originating from different fires).

• Also required is a clear description of what the lidars measure and how the intensive variables inform us about aerosol properties. Towards the end of the paper, for example, lidar ratio is explicitly referred to as 'absorption', but nowhere in the paper is there an explanation of why the authors make this connection (nor even clear references to that effect). On p.7 1.26-9 the paper simply states that this is so with no explanation. It would be better to provide a clear explanation in the Methods section of what the intensive variables tell us – that would then justify why they are used. There is discussion on p.4 of fresh and aged smoke which could be extended for this purpose. Bear in mind that there will be readers of this paper who are not intimately familiar with Raman lidars.

The paragraph from p 7, 126-29 was moved to Methodology section. The reference of Veselovskii et al 2020 was cited as well.

Unfortunately, we did not analyse the smoke particle size. The minimum dataset required to derive particle size (3 particles backscatter coefficients and 2 particle extinction coefficients) was not found for many of our measurements. We did not investigate either Aeronet column integrated particle size (this would have overload the amount of work). Veselovskii et al. (2020) show in their paper the increase of LR with imaginary part of refractive index for constant particle size.

In Introduction we mentioned:

The analysis is made using intensive parameters (referred to as IPs), which are independent of the aerosol load and are solely aerosol type dependent.

On the other hand, the extensive properties (aerosol backscatter and extinction coefficients) are related with the aerosol load. Intensive parameters are derived from two extensive parameters (see Appendix A).

We added the following sentence in the Methodology when first mentioning backscatter and extinction coefficients.

Particle backscatter and extinction coefficients represent extensive parameters and are related with the particle load.

The following sentence was added when mentioning Table 1 (Methodology).

The mean, median, minimum and maximum values of the intensive parameters for all of the stations providing at least one parameter (except Sofia station) are shown in Table 1. The number of available values for each variable is shown as well (# lines). Intensive parameters such as LR, EAE, BAE and PDR help identifying the aerosol type (e.g., Groß et al., 2013; Mylonaki et al, 2021).

The reference by Mylonaki et al was added to References:

Mylonaki, M., Giannakaki, E., Papayannis, A., Papanikolaou, C.-A., Komppula, M., Nicolae, D., Papagiannopoulos, N., Amodeo, A., Baars, H., and Soupiona, O.: Aerosol type

classification analysis using EARLINET multiwavelength and depolarization lidar observations, Atmos. Chem. Phys., 21, 2211–2227, https://doi.org/10.5194/acp-21-2211-2021, 2021.

In Introduction we cited several papers which studied fresh or aged smoke by analysing the smoke optical properties. In Part 1, in the Supplement are provided the plots with the intensive parameters found in the literature (Fig. S3) as well as their numerical data (Table S3). In general, the classification is based on smoke travel time. For long range transport smoke (e.g., from N America) one can be confident that the smoke is aged. For local fires, the delimitation based on travel distance is debatable. Some authors consider fresh smoke when travel distance is 1 day the most, others consider two days or even three days.

We added the following sentence regarding the aging in the discussion about fresh versus aged smoke:

Several authors report aged smoke when LR@532 > LR@355 (e.g., Mattis et al., 2003; Murayama et al., 2004; Müller et al., 2005; Sugimoto et al., 2010).

The following references were added:

Murayama, T., Müller, D., Wada, K., Shimizu, A., Sekiguchi, M., and Tsukamoto, T.: Characterization of Asian dust and Siberian smoke with multiwavelength Raman lidar over Tokyo, Japan in spring 2003, Geophys. Res. Lett., 31, L23103, https://doi.org/10.1029/2004GL021105, 2004.

Sugimoto, N., Tatarov, B., Shimizu, A., Matsui, I., and Nishizawa, T.: Optical Characteristics of Forest-Fire Smoke Observed with Two-Wavelength Mie-Scattering Lidars and a High-Spectral-Resolution Lidar over Japan, SOLA, 6, 093–096, https://doi.org/10.2151/sola.2010-024, 2010.

• The greatest flaw in the methodology of this paper is the use of single 10-day backtrajectories to infer the origin of the smoke layers measured. This is such a severe limitation that it could even be taken as grounds to reject the paper. Nowhere is there a discussion of the huge uncertainties in back-trajectories, or of the need for clusters of them to decide whether the flow is sufficiently non-dispersive to make them valid. Over 10 days in the troposphere most trajectories will be highly dispersive. This means that attributing source regions to the measurements is fraught with error. In addition, as the authors do recognise but do nothing about, the passage of a trajectory 9 km above a fire only implies a causal link if the fire plumes extended to 9 km, which in most cases they don't. I realise that this may well be an intractable problem, and that the results of this study may be novel enough to publish anyway, but the paper must discuss how the uncertainty in their source apportionment methodology affects the conclusions they draw. (Crudely, it makes any attempt to draw a distinction between different source regions null and void).

Due to the large amount of data (We performed 1036 Hysplit runs for 1901 layers corresponding to 960 time stamps.) we convened to use one single back-trajectory. Almost all the papers published on BB using lidars refer to 1 or 2-3 events (case studies). In such cases, sometimes the authors make use of ensemble trajectories (e.g., Janicka et al., 2017; Ortiz-Amescua et al., 2017; Ansmann et al., 2018; Vaughan et al, 2018). One trajectory is largely used in the lidar community such that the following studies we referenced: Wandinger et al., 2002; Balis et al., 2004; Heese et al., 2008; Mattis et al., 2008; Noh et al., 2009; Mariano et al., 2010; Giannakaki et al., 2010; Sugimoto et al., 2010; Tesche et al., 2011; Sicard et al., 2012; Baars et al., 2012; Nicolae et al., 2013; Pereira et al, 2014; Giannakaki et al., 2016;

Stachlevska et al., 2017 and Stachlevska et al., 2018. On the other hand, 10-day backtrajectory is used in the following studies: Heese et al., 2008; Mattis et al., 2010; Tesche et al., 2011; Ansmann et al., 2018; Haarig et al., 2018. Taking into account that we covered hundreds of events over 10 years measurements, we convened to use 10-days runs such that we covered both local and long-range transport.

In Part 1 we discussed about backtrajectories the following:

Section 4.3, Part 1:

It is worth mentioning that the Hysplit model does not provide the uncertainty. In order to get a possible uncertainty of an individual trajectory, a trajectory ensemble is suggested (Rolph et al., 2017). We may assume that high uncertainties in the air-mass location may occur particularly over long periods of time (e.g. 10 d), which in conjunction with a fire's location may mean a missed fire or a fire detection that was not contributing to the measurement. Drexler (https://www.arl.noaa.gov/hysplit/hysplit-frequently-asked-questions-faqs/faq-hg11/, last access: 26 November 2019) mentions that the uncertainty is between 15% and 30 %. On the other hand, FIRMS may miss some fires (especially in a cloudy atmosphere). According to Giglio et al. (2016), the collection 6 MODIS has a smaller commission error (false alarm) as compared with Collection 5 (1.2% versus 2.4% respectively). The probability of fire detection (regionally) increased by 3% in boreal North America while staying almost the same in regions such as Europe or northern Africa. We have been using fires with a confidence level larger than 70 %. We did not investigate the injection height based on FRP in order to estimate whether the smoke of a particular fire indeed reached the altitude of the back trajectory. We would like to emphasize that, due to the satellite's polar orbit, the same geographical location can be seen four times a day at the Equator and more times as the latitude increases (due to orbit overlap). Thus, we may miss a certain number of fires (which burn less than a few hours, between the two orbits). However, we may consider those shortlived fires to be insignificant in smoke production.

Summary and conclusions, Part 1 as well as in the current part:

Uncertainties in Hysplit back trajectories as well as in the FIRMS database are not considered.

We add the following sentence in Methodology:

As mentioned in part 1, high uncertainties in the air-mass location may occur over long periods of time of the air mass transport which in conjunction with the fire's location may mean that we missed a fire or we credited a fire as contributing to our measurement. Another source of uncertainty may come in the cases where the injection heigh did not reach the air-mass altitude (we did not investigate the injection height).

• Section 3.1 lines 20 - 30. A number of features in fig.1 are discussed here but it is not possible to relate them to the figure as that is organised by layer number not date (and for the first point discussed not even a date is given). Please point out exactly where in fig 1 the discussion is addressing in each case.

Please note that the representation versus date is not suitable as the time covers a large period and the events are usually located close each other around specific dates. Thus, the figure would have been overloaded and hard to read it for specific date.

We added the case # for the events we referred to:

Figure 1 shows the intensive parameters and layer altitudes measured during the LRT of smoke from North America (smoke originating in North America is shown in black and mixed smoke in blue). At a first glance, there is no evidence of a systematic difference

between the two categories. The mean (line), minimum and maximum (shaded areas) values from the literature are displayed in red (for the variables displayed by '\*') and green (for the variables displayed by 'o') corresponding to the references presented in Table 2. Compared to the values found in the rather limited existing literature for smoke originating in North America and measured over Europe (only tropospheric measurements were considered), we noted several IP values (especially for BAE@355/532) that fall outside of the range reported. The large value for the mixed smoke EAE (case # 70) may be due to the contribution of the local, fresh smoke. At a closer look, the large 'pure N America' EAE value (case # 9), recorded on 4 July 2013 in Thessaloniki in a layer at ~ 3.6 km altitude, corresponds to air masses reaching  $\sim 9 - 11$  km over the fires in North America. It is possible that the fires did not reach that altitude and thus the measurements for that layer may come from other sources. On the other hand, biomass burning particles can be found even in the lower stratosphere (e.g., Hu et al., 2019). The smallest EAE value (negative) (case # 49) may be due to dust contamination for a measurement performed in Granada on 19 August 2013 at 20:45 UTC, when fires in Portugal and North America were found along the backtrajectory. We also observe that mean PDR values are in general smaller if compared to the mean over the values reported in the literature. However, but still within the extreme values for smoke originating in North America (see Fig. 1 and Table 2). The minimum value reported for PDR@355 was 0.01±0.001 and for PDR@532 0.023±0.003 (Janicka et al., 2019). An EAE extreme value of -0.3 was reported by Haarig et al. (2018) but for stratospheric smoke.

• Section 3.1 lines 31 – p.6 l.3. The text here needs to be more quantitative and again refer exactly to the supporting evidence in fig.1. Terms like 'moderate', 'high' and 'low' are out of place here because the reader doesn't know the context within which these comparators sit. Furthermore, is there enough evidence to support sweeping generalisations like that on absorption, based on what looks like one event? The section also requires proper referencing to support the interpretations being presented.

The section where we specify these limits for low, medium, high absorption is moved to Section 2 (Methodology). However, we replaced moderate with medium, for consistency. References were added.

The PDR in Table 2 was not in [%]. We deleted [%]. We are sorry for mistake.

We added few details to further clarify the paragraph:

Overall, based on the mean values (Table 2, All), we observed for North America fire particles advected over Europe a medium absorption at 355 nm (LR@355 =  $46 \pm 16$ ) and a high absorption at 532 nm (LR@532 =  $66 \pm 32$ ) (CRLR > 1), with low depolarization at both wavelengths (< 0.04), relatively small EAE (apart from 2 isolated cases) - indication of big particles, slightly larger BAE@355/532 than BAE@532/1064. CRLR and EAE suggest the presence of aged particles, while BAE shows more backscatter for smaller wavelengths indicating small particles.

• The authors should carefully consider moving Fig 7 to the Supplement and removing the final paragraph of 4.2.1 which is descriptive and offers no interpretation of any value.

Figure 7 was moved to Supplement (as Figure S.5) and the paragraph was removed.

**Typos and minor corrections (of which there are many more than listed here)**

p.2 l.21 'indicate that climate change'

**corrected**

p.2 1.30 EARLINET provides remote-sensing lidar measurements not ground-based measurements. The lidars are on the ground but the measurements are not.

We replaced

provides high temporal and spatial resolution ground-based aerosol measurements, and represents a valuable tool

with:

provides high temporal and spatial resolution of aerosol measurements from ground-based lidars

p.3 l.6 origins

corrected

p.3 1.8 'This paper presents Part 2 of the investigation ......EARLINET, focussing on interpreting the results'

corrected

p.3 l.20 'averages over1 h'

corrected

p.3 1.22 'presence of smoke layers'

corrected

p.3 1.22-3 '.....stations, typically by means of......'

corrected

p.3 1.28 'described in detail in Part 1'

corrected

p.4 l.6 'middle points of a 1 km grid'

corrected

p.4 1.12 'criterion is presented in Table 2'

We guess you refer to line 10. We corrected.

p.4 1.13. Why do you explain BAE and PDR but not the other acronyms here? Either define

all of them or just refer the reader to Appendix A.

We changed the text

BAE represents the backscatter Ångström exponent and PDR represents the linear particle depolarization ratio (see Appendix A).

То

Please refer to Appendix A for LR, EAE, BAE and PDR description,

p.4 1.25 'which allowed the degree of oxidation in BB aerosol to be estimated'

corrected

p.5 l.14 and fig 1 caption 'layer altitudes'

corrected

p.5 l.16 and fig 1 caption 'the literature'

corrected

Fig 1 caption last line 'panel titles'

corrected

p.5 1.27 '....reported in the literature, but still within....'

corrected

p.5 1.30 delete 'the'

corrected

p.6 l.5 'with' instead of 'by'

corrected

p.6 1.6 'discuss stratospheric'

corrected

p.6 1.7 you can't have a single event lasting six months. What exactly did Baars et al do?

Baars et al. analysed stratospheric aerosol in EARLINET from August 2017 to January 2018. Please see https://acp.copernicus.org/articles/19/15183/2019/.

"Enormous amounts of smoke were injected into the upper troposphere and lower stratosphere over fire areas in western Canada on 12 August 2017 during strong thunderstorm–pyrocumulonimbus activity. The stratospheric fire plumes spread over the entire Northern Hemisphere in the following weeks and months."

p.6 1.9 you abruptly transition from previous studies to results from this study with no

explanation. You need to explain to the reader that you are now presenting your own results.

We guess you mention line 15 where we start to talk about our results. We added at the beginning of line 15:

In the present study, the locations of the fires...

p.6 l.16. Replace the sentence 'For a straightforward comparison, we reproduce the figure for the South-East region from Part 1' by '(Note that the figures for the south-east region

are the same as Figure 11 in Part 1)'

corrected

p.6 l.18 delete 'a number of'

corrected

p.6 l.27 'Eastern Europe', also delete 'region'

corrected

p.6 1.30 replace 'contained a' with 'observed', and omit 'the' before Cabauw.

corrected

p.7 l.2, 4 and 6 'Eastern Europe', also 'Southern' on l.6

corrected

p.7 1.8 'Events where small particles are transported in the boundary layer from these regions to North-West Europe.....'

corrected

p.7 l.11 Arctic (not arctic)

corrected

p.7 1.23 'results for the South-East region'

corrected

p.7 1.26-29 - this paragraph is out of place and should form part of an expanded

Introduction or be included in the methodology section when lidar ratio is introduced.

There should be more references to the link between absorption and lidar ratio as well.

We moved the paragraph in Methodology, after discussing LR and before the last paragraph of the section. We cite also the reference of Veselovskii et al, 2020.

p.8 l.2 'LR@355 being larger...' (the verb in this sentence is 'indicates'). 'From the EU

region' on 1.3

corrected

p.8 1.9 'the supplement'

corrected

p.8 1.15 'which are larger'

corrected

p.8 l.17 'between the NA and EUNA....', then 'the EUNA' on the next line

corrected

p.8 l.22 'the South-West'

corrected

p.8 1.23 'the EU and AF'

corrected

p.8 l.24 'indicate a large'

corrected

p.8 1.25 'Similar, large, values ... for the NA....'

corrected

p.8 1.26 'the EUAF'

corrected

p.8 1.27 'the AF'

corrected

p.8 1.28 'the North-East'

corrected p.8 1.31 'the EU' corrected p.9 1.1 'the EUAF' corrected p.9 1.4 'the EU' and 'the NA' corrected

p.9 1.7 Replace 'We perform the analysis based on the mean IP values as a function of continental source region. We consider analysing the scatter plots between the different CRs and EAE, where, for each scatter plot, the mean values correspond to the same measurements. Still, different scatter plots can refer to slightly different sets of measurements.' With ''We present analyses of mean IP values as a function of continental source region by means of scatter plots between the different CRs and EAE'. I didn't understand what you meant by 'for each scatter plot, the mean values correspond to the same measurements.'

We replaced as suggested. The statement 'for each scatter plot, the mean values correspond to the same measurements' means that each of the mean values on x-axis and y-axis are obtained for the same set of measurements. In other words, we have 5 measurements where we can compute the mean value both on x-axis and y-axis. We tried to make it clear because some may use other option such as the mean value on x-axis is over 5 measurements and the mean value on y-axis is over 10 measurements (in this case, the means do not correspond to the same measurement dataset).

One can see that the plots d)-f) contain more data. Thus, we specify Still, different scatter plots can refer to slightly different sets of measurements.

Fig 6 – why are there two 'literature means' on panels a-c and several on panels d-f?

First, we corrected the wrong reference for literature values (p 9, 1 17). It is Table S3, Part 1 instead of Table S1, Part 1. We apologize. That table contains all the values found in literature (46 citations – see Table S4 in Part 1) for LR@355, LR@532, EAE, BAE@532/355, BAE@1064/532, PDR@355, PDR@532. We cited whatever it was available, e.g., mean values, min and/or max values and associated errors. The number of points in Fig. 6 represent all the mean values available in the corresponding pair (e.g., CRBAE and CRPDR, EAE and CRPDR etc). Thus, we found two pairs of values for the panels a) and b), three pairs of values for the panel c), nine pair of values for d) and seven pair of values for e) and f).

We did not perform an average over all literature values found. We picked the available mean values (in pairs) and added them to the plots. Maybe this was the confusion.

p.9 1.23 'except for'. Also the lowest EAE is <0, not <0.5. The sentence degenerates into confusion. CRPDR < 1 means (not indicates) that PDR355 > PDR532 as that is how it is defined. It may indicate aged particles except that the next clause (1.24) contradicts this and says that PDR532 can be greater for either fresh or aged smoke. (This is why a clear

literature survey earlier in the paper is so important). What are you trying to say?

The smallest value for EAE (seen an panels b, d and f) is 0.25. Please do not confuse with the literature values (empty red circles) which have values <0 indeed. We are sorry for the confusion. We rephrased:

Based on panel b), we observe that except for the case with low EAE (<0.5) and CRPDR<1 which indicates aged particles and smaller depolarization at 355 nm, the depolarization at 532nm can be higher for either fresh or aged smoke.

As:

Based on panel b), we observe that except for the case with low EAE (<0.5) and CRPDR<1 (which indicates aged particles and smaller depolarization at 532 nm than at 355 nm) the depolarization at 532nm increases from aged to fresh smoke (EAE >1.4). The decrease of depolarization ratio at 532 nm with travel time was reported by Nisantzi et al. (2014).

The reference by Nisantzi was added:

Nisantzi, A., Mamouri, R. E., Ansmann, A., and Hadjimitsis, D.: Injection of mineral dust into the free troposphere during fire events observed with polarization lidar at Limassol, Cyprus, Atmos. Chem. Phys., 14, 12155–12165, https://doi.org/10.5194/acp-14-12155-2014, 2014.

We tried to show that depolarization at 532 nm < depolarization at 355 nm as we go from fresh to aged smoke, i.e., CRPDR decreases towards aged smoke.

p.9 l.24 Datasets cannot be statistically significant – what correlation exactly does this refer to?

The correlation refers to the trend between EAE and CRPDR (increase of EAE with increase of CRPDR). We rephrased:

The dataset is not statistically significant, but increased number of samples in future studies is expected to reveal the statistical significance of this correlation.

As:

The dataset is not statistically significant, but increased number of samples in future studies is expected to reveal a statistically significant correlation.

p.9 l.25-6 'From fig 6a, a slight.....'

corrected

p.10 1.9 'the EU'

corrected

p.10 l.11 'a result'

corrected

p.101.12 'the EUNA', 'the EUAS'

corrected

p.10 1.32 'the NA'

corrected

p.10 1.33 'the EUNA'

corrected

p.11 l.1 'the local (EU) contribution determines whether ...'

corrected

p.11 1.3 'the EU' 'particle size'

corrected

p.11 l. 5 'the EUAF ..... the EU'

corrected

p.11 1.13 'the EUAF'

corrected

p.12 1.10 and 11 ')' after America.

corrected

p.12 1.13 'The analysis..' - the sentence does not make sense and needs redrafting

the sentence:

The analysis of the scatter plots revealed correlated with the increase of smoke travel time (corresponding to aging), CRLR and CRBAE increase while EAE and CRPDR decrease.

was rephrased as:

The analysis of the scatter plots revealed correlations with the increase of smoke travel time (corresponding to aging). Thus, CRLR and CRBAE increase while EAE and CRPDR decrease.

p.12 l.15 change 'helping identifying the smoke' to 'helps to identify smoke'

corrected

p.12 l. 30 'measurement regions'

corrected

p.13 l.2 'the ACTRIS'

corrected

p.13 l.3 'applied to'

corrected

p.13 1.4 'providing more ... datasets' or 'providing a more .... dataset'

the sentence was changed to:

...the presented methodology will be applied to a larger dataset (more automatic lidar systems expected) providing a more complete (3 backscatter + 2 extinction + 1-3 depolarization) dataset with enhanced quality control procedures.

p.13 1.8. Remove double .. 'shows the potential of the approach described'

corrected

p.13 1.10 'results obtained' not 'obtained results'

corrected

p.13 1.13 remove the sentence 'This extension...' It is superfluous and also flawed grammatically deleted